# Single-cell profiling defines the prognostic benefit of CD39<sup>high</sup> tissue resident memory CD8+ T cells in luminal-like breast cancer

Agnese Losurdo[1,2,9 ✉], Caterina Scirgolea [1,9], Giorgia Alvisi[1], Jolanda Brummelman [1], Valentina Errico[3], Luca Di Tommaso[4,5], Karolina Pilipow[1], Federico Simone Colombo [6], Bethania Fernandes [4], Clelia Peano [7,8], Alberto Testori[3], Corrado Tinterri[3], Massimo Roncalli[4,5], Armando Santoro [2,5], Emilia Maria Cristina Mazza [1] & Enrico Lugli [1 ✉]

Luminal-like breast cancer (BC) constitutes the majority of BC subtypes, but, differently from highly aggressive triple negative BC, is poorly infiltrated by the immune system. The quality of the immune infiltrate in luminal-like BCs has been poorly studied, thereby limiting further investigation of immunotherapeutic strategies. By using high-dimensional single-cell technologies, we identify heterogeneous behavior within the tissue-resident memory CD8+ T (Trm) cells infiltrating luminal-like tumors. A subset of CD127− CD39<sup>hi</sup> Trm cells, preferentially present in the tumor compared to the adjacent normal breast tissue or peripheral blood, retains enhanced degranulation capacity compared to the CD127+ CD39<sup>lo</sup> Trm counterpart ex vivo, and is specifically associated with positive prognosis. Nevertheless, such prognostic benefit is lost in the presence of highly-suppressive CCR8<sup>hi</sup> ICOS<sup>hi</sup> IRF4+ effector Tregs. Thus, combinatorial strategies aiming at boosting Trm function and infiltration while relieving from Treg-mediated immunosuppression should be investigated to achieve proper tumor control in luminal-like BCs.

[1] Laboratory of Translational Immunology, IRCCS Humanitas Research Hospital, Rozzano, Milan, Italy. [2] Medical Oncology and Hematology Unit, IRCCS Humanitas Research Hospital, Rozzano, Milan, Italy. [3] Breast Surgery Unit, IRCCS Humanitas Research Hospital, Rozzano, Milan, Italy. [4] Department of Pathology, IRCCS Humanitas Research Hospital, Rozzano, Milan, Italy. [5] Department of Biomedical Sciences, Humanitas University, Pieve Emanuele, Milan, Italy. [6] Humanitas Flow Cytometry Core, IRCCS Humanitas Research Hospital, Rozzano, Milan, Italy. [7] Genomic Unit, IRCCS Humanitas Research Hospital, Rozzano, Milan, Italy. [8] Institute of Genetic and Biomedical Research, UoS Milan, National Research Council, Rozzano, Milan, Italy. [9] These authors contributed equally: Agnese Losurdo, Caterina Scirgolea. ✉email: agnese.losurdo@cancercenter.humanitas.it; enrico.lugli@humanitasresearch.it

The interplay between the immune system and cancer cells has been recognized as a hallmark of cancer, involved in its pathogenesis, growth, and resistance to medical treatments[1]. Immunotherapeutic strategies, mainly using mono-clonal antibodies (mAbs), act by blocking the activation of immune checkpoint proteins such as the programmed death-1 (PD-1) and its ligand, PD-L1, that behave as coinhibitory factors, thereby halting or limiting the development and functionality of T-cell responses. Breast cancer (BC) has long been considered a nonimmunogenic cancer type. However, it is a very hetero-geneous disease, with different molecular subtypes harboring different biological characteristics, extent of immune infiltration, and prognostic significance[2]. In routine clinical practice, luminal-like, HER2-enriched and triple negative (TN) BCs are recognized based on estrogen receptor (ER) and progesterone receptor status and HER2 protein expression. TNBCs and HER2-overexpressing tumors are the more aggressive subtypes and display an intrin-sically higher gene mutation rate and tumor-infiltrating lym-phocyte (TIL) abundance compared to luminal-like tumors, which are generally referred to as immunologically cold[3,4].

Seminal studies revealed that the abundance of TILs as a whole define individuals with better prognosis in TNBC but not in luminal-like BCs treated with adjuvant chemotherapy, suggesting a limited role of the immune system in the antitumor immune response in luminal-like BC[5–8]. However, recent data have shown that cyclin-dependent kinase 4/6 (CDK4/6) inhibitors, now standard of care for the treatment of metastatic luminal-like tumors, promote antitumor immunity in preclinical models by increasing tumor antigen presentation, IFN-γ production, and CD8+ T-cell expansion, and by inhibiting CD4+ T regulatory (Treg) cell proliferation[9]. Moreover, CDK4/6 inhibitors and anti-PD-1 combination therapy resulted in enhanced and durable tumor shrinkage[9], overall suggesting that T cells might play a nonredundant role in antitumor immunity also in luminal-like BC subtypes.

Recent investigations by high-dimensional single-cell technol-ogies such as single-cell RNA sequencing (scRNA-seq) or by cytometry by time of flight elucidated in part the complexity of the BC immune infiltrate. A recent, comprehensive single-cell analysis of 144 cases defined that BC-associated immune cells are mostly T and myeloid cells, with natural killer (NK) cells, B cells, and granulocytes being less represented[10]. Among CD8+ T cells, the majority bear CCR7−CD45RA− effector memory T (Tem) and CCR7−CD45RA+ effector memory reexpressing CD45RA (Temra) phenotypes[10–12], where the former preferentially expressed increased levels of PD-1, T-cell immunoreceptor with Ig and ITIM domains (TIGIT), and 2B4, but not T-cell immu-noglobulin mucin-3 (TIM-3) and lymphocyte-activation gene 3 (LAG-3) inhibitory receptors compared to other BC-infiltrating CD8+ T-cell subsets[13]. Specifically, PD-1[high] CD8+ cells that also expressed the inhibitory receptor cytotoxic T-lymphocyte antigen-4 (CTLA-4) were found to be enriched in tumor tissue compared to normal breast and to be represented in both ER− and highly proliferative ER+ malignancies[10]. Nevertheless, these PD-1+ CD8+ BC-infiltrating TILs, which are generally con-sidered to be exhausted in the tumor microenvironment (TME), were found to retain some degree of polyfunctionality, such as cytokine production capacity and degranulation, when compared to those TILs infiltrating melanoma, thereby suggesting a differ-ential regulation of these cells in BC[13]. Subsequent studies have shown that, within the broadly defined Tem-phenotype cells, those T cells expressing the tissue-resident memory (Trm) mar-kers CD103+ and/or CD69+ display antigen specificities that are different from circulating Tem cells, as evidenced by non-overlapping T-cell receptor repertoires[14]. In tumors such as non-small cell lung cancer (NSCLC), head and neck squamous cell carcinoma, and colorectal cancer, Trm cells preferentially expressed CD39, a surrogate marker of T cells recognizing tumor-specific antigens[15,16]. In line with their tumor reactivity, the abundance of Trm in tumor stroma was associated with better overall survival (OS) and disease-free survival in TNBC[12] and in NSCLC[17,18]. Thus, to date, many studies have investigated the quality of the immune infiltrate of TNBC, the most aggressive subtype of BC, while that of the more indolent luminal-like subtype remains poorly characterized, thereby leaving it unclear whether specific immune characteristics play a role in disease progression.

Here, we applied high-dimensional single-cell profiling to dozens of luminal-like BCs to investigate the complexity of T-cell phenotypes and defined those T-cell characteristics that are associated with progression of the disease. Specifically, the extent of infiltration of a small subset of putatively antigen-specific CD127− CD39[hi] CD8+ Trm cells identified patients with improved survival, especially when associated with low abun-dance of activated CCR8[hi] ICOS[hi] Tregs. These results give important hints on the potential immunogenicity of hormone receptor-positive disease, suggesting that immunotherapeutic manipulations may be beneficial also in this BC subtype.

## Results

**BC immune infiltrates are enriched in Trm cells.** We initially took advantage of publicly available data from scRNA-seq analyses[11] to investigate the complexity of BC-infiltrating T cells, so to identify signature markers of discrete subpopula-tions that could subsequently guide T-cell profiling of a large number of patients' specimens by high-dimensional flow cyto-metry. Reanalysis of 3637 T cells sorted in silico from eight patients (four luminal-like, three TNBC and one HER2-enriched) on the basis of CD3E expression identified five and six clusters of CD4+ and CD8+ T cells, respectively (Fig. 1a). The majority of CD4+ T cells (C0) overexpressed KLF2, CCR7, SELL, and JUNB, thereby identifying early differentiated memory T cells. We identified additional clusters of memory, yet not terminally dif-ferentiated CD4+ T cells, overexpressing IL7R and MALAT1 (C1), and clusters of cytotoxic/effector-like cells, overexpressing GZMA, GZMK, and CXCR6 (C2), or GZMB, GNLY, HLA.DRA and IFNG (C4). An additional cluster, C3, overexpressed FOXP3, TIGIT, ICOS, and CTLA4, thereby identifying regulatory CD4+ T cells (Treg; Fig. 1a, CD4). As far as CD8+ T cells were con-cerned, we revealed an abundant cluster of memory T cells, C1, overexpressing IL7R, KLRC1 (encoding NKG2A), and KLRB1 (encoding CD161), two clusters of cytotoxic/effector-like CD8+ T cells overexpressing GZMK and GZMA (C0) or GZMB, GZMH, GNLY, and PRF1 (C3), and a Trm cluster, C4, overexpressing ITGAE (encoding CD103) and CD69 (Fig. 1a, CD8). C2 and C5 could not be defined with precision on the basis of their gene expression.

We next designed a scRNA-seq-guided 27-parameter flow cytometry panel including signature markers informative of the differentiation, activation, proliferation, and exhaustion status of BC TILs (Supplementary Table 1). We profiled millions of single cells from the tumor, normal breast tissue, and peripheral blood of 54 treatment-naive BC patients surgically treated at our institution. Our cohort of consecutive patients mirrors the well-known epidemiology, with 37 (69%) luminal-like HER2 negative, 10 (18%) luminal-like HER2-overexpressing, 2 (4%) hormone receptor negative HER2-overexpressing (HER2- enriched), and 5 (9%) TNBCs. Detailed patients' characteristics are summarized in Supplementary Table 2. By applying the unsupervised clustering algorithm PhenoGraph, we identified 12 and 10 different clusters of CD4+ and CD8+ T cells, respectively (Fig. 1b; see "Methods").

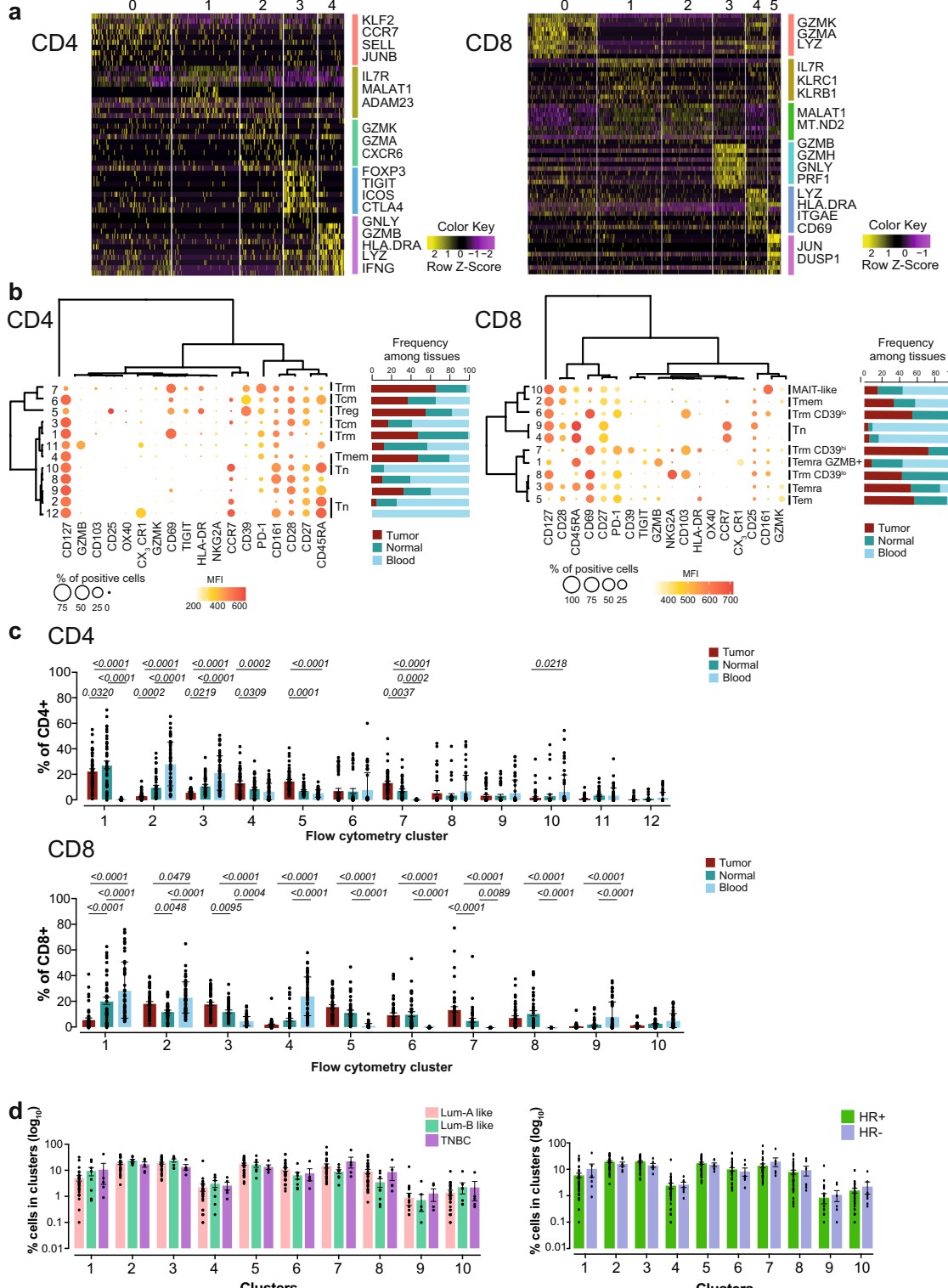

**Fig. 1 Breast cancer (BC) immune infiltrates are enriched in Trm cells. a** Heatmaps showing single-cell gene expression by T-cell clusters from tumor-infiltrating CD3+ cells from BC patients ($n = 475$ cells for CD4+ and 1167 cells for CD8+). Signature genes of the specific clusters are indicated. **b** Balloon plot maps showing the percent expression and the median fluorescence intensity (MFI) of specific markers (columns) in discrete PhenoGraph clusters (rows) as identified from the analysis of 54 patients. Hierarchical metaclustering grouped markers and clusters with similar immunophenotypes. Bar plots show the percent frequency of each cluster among different tissues (tumor, normal tissue, and peripheral blood). Treg CD4+ regulatory T cells, Trm tissue-resident memory, Tcm central memory, Temra effector memory reexpressing CD45RA, MAIT mucosal-associated invariant T, Tem effector memory. **c** Bar plots showing the frequency depicted as mean ±SEM of CD4+ and CD8+ T-cell PhenoGraph clusters in peripheral blood, normal breast tissue, and tumor samples from data obtained in **b**. Numbers indicate the exact *p* values for tumor vs. peripheral blood or normal tissue samples; two-way ANOVA with Bonferroni post hoc test. **d** As in **c**, but among luminal A-like, luminal B-like, and triple negative breast cancer (TNBC) biological subtypes and among subtypes according to positive or negative expression of hormone receptors (HR).

Visualization of single-cell clusters using UMAP (Supplementary Fig. 1a) and metaclustering of PhenoGraph clusters (Supplementary Fig. 1b) revealed pronounced differences in T-cell phenotypes among the three different specimens, both for CD4+ and CD8+ T-cell populations. As previously observed for other types of cancer in the tumor compared to the blood[19,20], we noticed the decrease in naive T cells (Tn; mainly represented by C2 for CD4+ and C4 for CD8+), CD4+ central memory T (Tcm) cells (CD4+ C3), CD27+ CD28+ early memory CD8+ T cell (Tmem; CD8+ C2), and GZMB+ CD27− CD28− CX$_3$CR1$^{dull}$ CD8+ Temra cells (CD8+ C1), accompanied by the increase in HLA-DR$^{hi}$ CD39$^{hi}$ Tregs (CD4+ C5), as well as effector CD8+ T cells featuring CD69, the inhibitory receptors PD-1 and, in part, GZMK (CD8+ C3 and C5) (Fig. 1b, c). These clusters also lacked the killer molecule GZMB (Fig. 1b, c). Moreover, the tumor as well as the adjacent, nontumoral tissue displayed the increased presence of T cells featuring the Trm markers CD69 and CD103 (CD8+ C6, C7, and C8) or CD69 only (CD4+ C1 and C7), hereafter referred to as Trm (Fig. 1b, c). Specifically, PhenoGraph clustering identified phenotypic heterogeneity in the CD8+ Trm population not identified by scRNA-seq, according to which C6 and C8 shared a similar identity except for the expression of the inhibitory receptor NKG2A, while a third, more diverse subset of Trm cells, C7, could be distinguished on the basis of CD39 positivity, slightly increased levels of HLA-DR and absence of CD127 compared to C6 and C8 subsets of Trm (Fig. 1b). Of note, CD39 expression has been recently linked to CD8+ T-cell reactivity to tumors[15,16]. The relative abundance of T-cell clusters was similar in luminal-like A and B and TNBC or between BCs with different endocrine dependence (Fig. 1d), thereby revealing equivalent patterns of T-cell differentiation among the different BC subtypes.

**CD8+ Trm cells are phenotypically and functionally heterogeneous in BC.** Recent data uncovered a significant association between the infiltration of Trm cells and good prognosis in basal-like BC[21] and TNBC[12]. Our high-dimensional single-cell analysis identified putative heterogeneity within the CD8+ Trm compartment in BC on the basis of CD127 and CD39 expression. Analysis by UMAP (Fig. 2a) indeed revealed that, on average, CD39 was not expressed with CD127 within CD8+ Trm cells, rather with HLA-DR, PD-1, CD27, and, at a lesser extent, TIGIT. Manual gating analysis of CD8+ Trm subsets on the basis of CD127 and CD39 expression identified three major subsets of CD127+ CD39$^{lo}$, CD127− CD39$^{lo}$, and CD127− CD39$^{hi}$ Trm (Supplementary Fig. 2a), on which the expression of different markers was quantified by median fluorescence intensity (MFI) by flow cytometry (Fig. 2b). We observed increased HLA-DR activation marker and PD-1 and TIGIT inhibitory receptors expression by CD127− CD39$^{hi}$ Trm compared to other subsets. These cells also expressed relatively high levels of GZMB and low levels of GZMK. Conversely, the inhibitory receptor NKG2A, known to be expressed on intratumoral CD8+ CD103+ effector T cells and linked to progression in head and-neck squamous cell carcinoma patients[22], was more prominent in the CD127+ CD39$^{lo}$ Trm subpopulation (Fig. 2b). We next quantified the amount of the three subsets of Trm by manual gating of flow cytometry data in our cohort of patients and found that their frequency among CD8+ increased in the tumor compared to the blood. In the case of CD127− CD39$^{lo}$ and CD127− CD39$^{hi}$ Trm, the frequency was also higher than in the adjacent normal tissue (Fig. 2c). CD127− CD39$^{hi}$ Trm could also be detected in lymph nodes collected at the time of primary surgery (Supplementary Fig. 2b) and preferentially infiltrated metastatic, tumor-draining lymph nodes compared to tumor-free lymph nodes (Fig. 2d),

suggesting that CD39 identifies putative tumor-reactive CD8+ Trm cells also in BC similar to other types of human tumors[15,16]. We next investigated the effector functional capacity of the CD8+ Trm subsets identified by CD127 and CD39 expression following nonspecific stimulation with phorbol 12-myristate 13-acetate (PMA) and ionomycin in vitro and measurement of effector molecules production by flow cytometry[19,20,23]. In line with the enhanced expression of the killer molecule GZMB ex vivo, we observed that CD127− CD39$^{hi}$ Trm stained more frequently for CD107a ($p \leq 0.05$) than CD127+ CD39$^{lo}$ Trm after stimulation, while IFN-γ and TNF were equally produced among the three subsets (Fig. 2e, f). Conversely, expression of IL-2 seemed more evident in CD127+ CD39$^{lo}$ Trm compared to CD127− CD39$^{hi}$ Trm ($p \leq 0.05$) (Fig. 2e, f). Thus, in comparison to normal breast tissue and blood, tumors are enriched in subsets of Trm cells, defined according to CD127 and CD39 expression, that produce a diverse combination of effector molecules upon PMA/ionomycin stimulation.

**CD8+ Trm subsets and their role in BC prognosis.** We next focused on CD127− CD39$^{hi}$ Trm, given their possible implication in the antitumor immune response as described in other human tumors[15,16], and compared them to the most diverse CD127+ CD39$^{lo}$ Trm cells. We thus isolated the two subsets by fluorescence-activated cell sorting (FACS; the sorting gating strategy is depicted in Supplementary Fig. 3a) and performed bulk RNA-seq to gain more insights on their molecular features. When taking into account only protein coding genes, we found that the two Trm subsets were relatively different at the transcriptomic level, and identified 183 differentially expressed genes, 117 of which were upregulated, while 66 were downregulated in CD127+ CD39$^{lo}$ compared to CD127− CD39$^{hi}$ Trm cells ($q < 0.05$; Fig. 3a and Supplementary Table 3). The former preferentially expressed genes related to early memory differentiation such as *IL7R* (encoding CD127), as expected from the sorting strategy, *CCR7* and *RUNX2*, as well as the additional immune related genes *STAT4*, *KLRD1* (encoding CD94, a molecular adapter of NKG2A, in turn preferentially expressed by CD127+ Trm cells at the protein level), *RORA*, and *DUSP2*. Instead, the latter preferentially expressed *MYB*, regulating survival programs of memory T cells and promoting their antitumor response[24], *CDH1*, linked to T-cell tissue residency[25], and *SOX4* and *IGF1R*, both involved in T-cell maturation[26,27]. Gene set enrichment analysis (GSEA) revealed that CD127+ CD39$^{lo}$ Trm cells were characterized, among others, by gene signatures related to oxidative phosphorylation, fatty acids metabolism, and p53 mediated hypoxia, which have been demonstrated to be linked to defective anti-tumor reactivity of T cells[28–30] (Fig. 3b), whereas the CD127− CD39$^{hi}$ Trm cells were enriched in signatures related to interaction with the extracellular matrix, and sodium and potassium channel activity[31–33] (Fig. 3b). We next employed these specific gene signatures to investigate the prognostic impact of CD127+ CD39$^{lo}$ and CD127− CD39$^{hi}$ subsets of Trm cells on BC prognosis, as defined in the Molecular Taxonomy of Breast Cancer International Consortium (METABRIC) dataset[34]. Overall, increased levels of *ITGAE* expression, reflecting Trm infiltration, correlated with slightly improved OS in luminal-like BC (Supplementary Fig. 3b), in line with previous data in TNBC[12] and NSCLC[17,18]. Specifically, an increased expression level of the CD127− CD39$^{hi}$, but not of the CD127+ CD39$^{lo}$ Trm cell signatures significantly correlated with better OS ($p = 0.0029$; Fig. 3c). The favorable prognostic effect of the CD127− CD39$^{hi}$ signature was confirmed also when taking into account only luminal-like tumors ($p < 0.0001$; Fig. 3d).

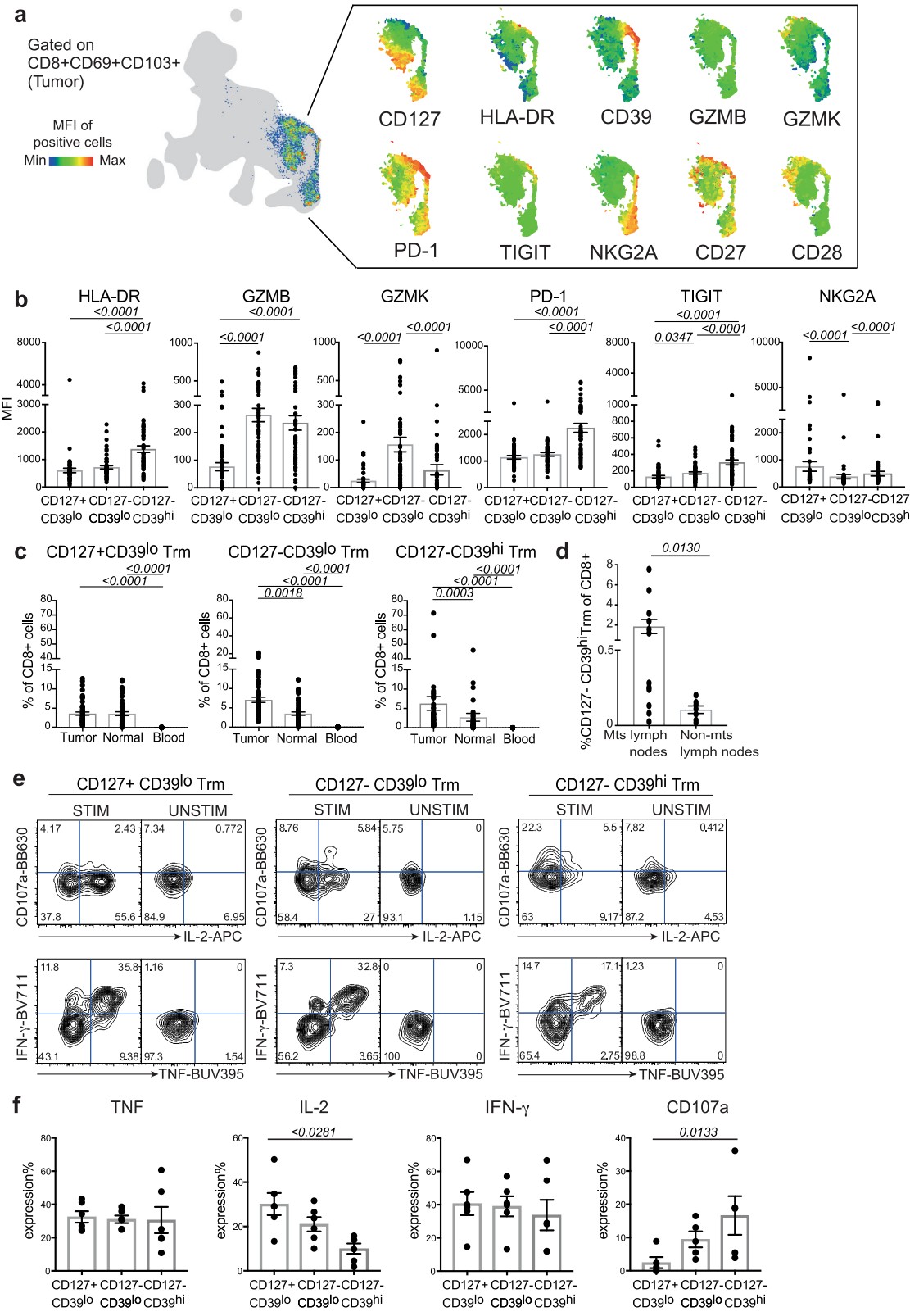

Similar data were obtained when analyzing only BC-specific mortality ($p < 0.0001$; Supplementary Fig. 3c). As expected, the prognostic benefit of CD127− CD39[hi] Trm cell infiltration correlated with the abundance of CD8+ T cells, evaluated as *CD8A* expression, in luminal-like BC tumors ($p < 0.0001$; Supplementary Fig. 3d).

**Abundance of CCR8[hi] ICOS[hi] IRF4+ effector Tregs hinders the prognostic benefit of Trm cells**. The tumor ecosystem is composed by several immune populations that can promote or inhibit antitumor immune responses. To define a possible interplay between subsets of Trm cells and other T cells in tumors, we performed a Pearson's correlation analysis of the abundance of

**Fig. 2 CD8+ Trm cells are phenotypically and functionally heterogeneous in BC. a** UMAP representation of the relative expression of selected markers analyzed by flow cytometry by manually gated CD8+ CD69+ CD103+ Trm cells concatenated from tumor samples ($n = 49$ biologically independent samples). The gray area refers to the total CD8+ T-cell subpopulation. **b** Mean ± SEM summary of the MFI of different activation, coinhibitory and costimulatory markers by CD127+ CD39lo, CD127− CD39lo, and CD127− CD39hi Trm subpopulations from tumor samples ($n = 54$ biologically independent samples), as defined by flow cytometry. Numbers indicate the exact $p$ values; one-way ANOVA with Friedman test. **c** Bar plots showing the frequency depicted as mean ± SEM of CD127+ CD39lo, CD127− CD39lo, and CD127− CD39hi in CD8+ cells among different tissue compartments (tumor, normal breast, and peripheral blood; $n = 54$ biologically independent samples). Numbers indicate the exact $p$ values for tumor vs. peripheral blood or normal tissue samples; one-way ANOVA with Friedman test. **d** Mean ± SEM summary of the frequency of CD127− CD39hi Trm among total CD8+ cells in metastatic (mts; $n = 12$ biologically independent samples) and nonmetastatic (non-mts; $n = 7$ biologically independent samples) lymph nodes (LN). Numbers indicate the exact $p$ value; Mann–Whitney test. Data were obtained in one experiment. **e** Flow cytometry plots from a representative patient showing the production of CD107a, IL-2, TNF, and IFN-γ by BC tumor-infiltrating CD8+ Trm subsets in response to PMA/ionomycin stimulation (STIM) in vitro. Numbers in the histograms indicate the percentage of positive cells identified by the gates. UNSTIM unstimulated control. **f** Mean ± SEM summary of the frequency of effector molecules expression from **e** ($n = 6$ biologically independent samples; performed in two independent experiments). Numbers indicate the exact $p$ value; one-way ANOVA with Friedman test.

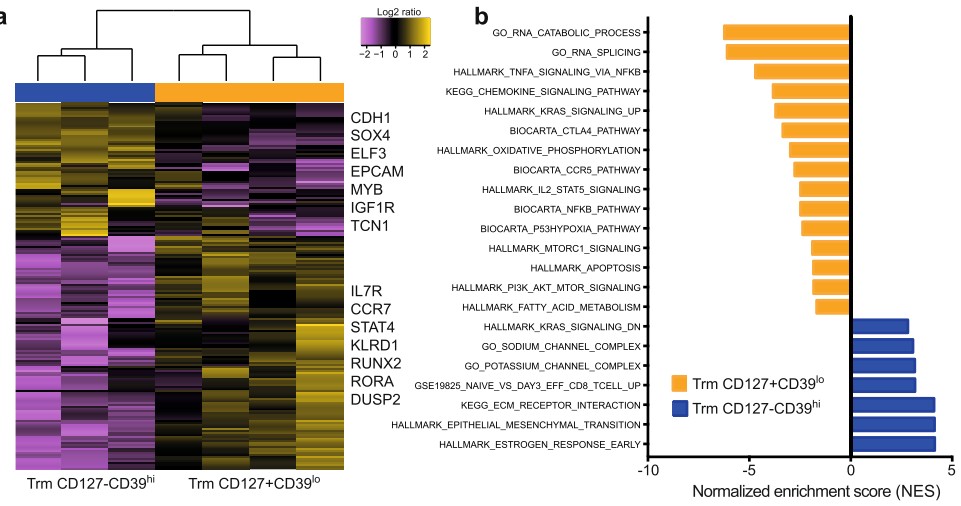

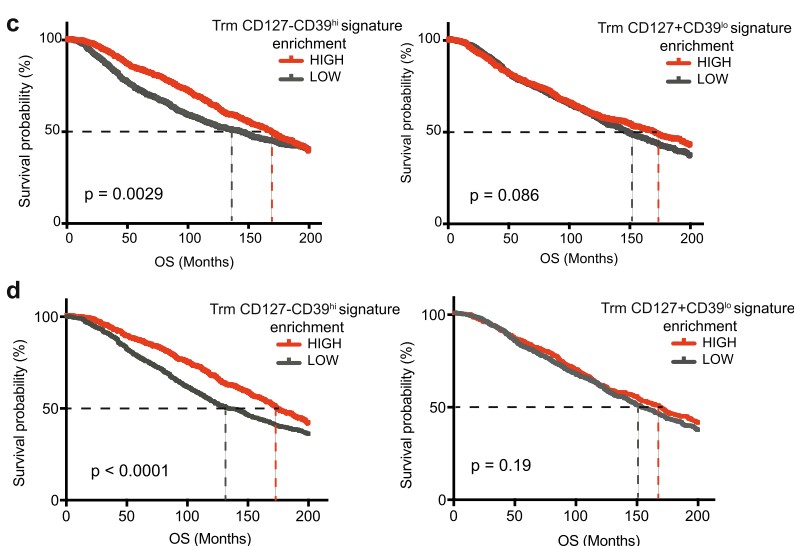

**Fig. 3 CD127− CD39hi Trm cells define a signature of increased survival in BC. a** Heatmap of differentially expressed genes (FDR < 0.05) between FACS-sorted CD127− CD39hi ($n = 3$) vs. CD127+ CD39lo ($n = 4$) CD8+ Trm cells subsets from tumor samples ($n = 6$ biologically independent samples), as obtained by RNA-seq. Selected differentially expressed genes are indicated. **b** Bar plot of manually curated signatures that differed significantly (adjusted $p$ value < 0.05) as obtained by gene set enrichment analysis (GSEA) of RNA-seq data in **a**. Kaplan–Meier overall survival (OS) curves in the METABRIC dataset for all BC patients (**c**; $n = 1894$) and for luminal-like tumors only (**d**; $n = 1436$). The mean $z$-score value was used to classify tumor samples into LOW and HIGH expression groups. $p$ values were calculated by applying the log-rank (Mantel–Cox) test. Dotted lines indicate the time at which 50% of the cohort was still free of the event.

cell clusters identified by high-dimensional flow cytometric analysis from Fig. 1b. We revealed that the frequency of the flow cytometry cluster C7 of CD8+, featuring CD127− CD39hi CD8+ Trm cells, was directly correlated with that of C5 of CD4+ featuring Tregs (Supplementary Fig. 4a). We confirmed the correlation by manually gated flow cytometry data (Supplementary Fig. 4b). These data suggest a possible relationship in the TME between these two populations of cells. Tregs in tumors are characterized by substantial heterogeneity, involving quiescent and activated subsets. The latter feature the expression of effector and activation markers, are more suppressive than quiescent Tregs, and correlate with worse prognosis in NSCLC, melanoma, and hepatocellular carcinoma[35]. To gain more insights on the heterogeneity in BC, CD4+ Tregs were isolated as CD25+ CD127− by manual gating of flow cytometry data from the three different tissue sites of the entire cohort (representative individuals are exemplified in Supplementary Fig. 4c), and further reclustered by PhenoGraph (Fig. 4a, b). As CD25 and CD127 expression in tumors could be modulated by cell activation and inflammation, we confirmed that the majority of intratumoral CD25+ CD127− T cells also expressed high levels of FOXP3 compared to CD25− CD127+ conventional (Tconv) and CD25− CD127− effector T (Teff) cells (Fig. 4c), both in terms of percentage and MFI of protein expression by flow cytometry (Fig. 4d). Additional markers previously described in intratumoral Tregs, including CD39, HLA-DR, and IRF4, were overexpressed by BC Tregs compared to Tconv and Teff, indicating consistency of these phenotypes among the different patients (Fig. 4d). PhenoGraph analysis informed on the differential abundance of seven clusters: CD45RA+ CCR7+ CD28+ CD27+ naive-like (C6) and CD45RA− CCR7+/− CD28+ early memory-like Tregs lacking PD-1, CD69, and HLA-DR (C2–C4) were more abundant in the blood, whereas subsets of CD45RA− CCR7dull CD28+ CD27+ Tregs, featured predominantly in C1 and C5, less so in C7, were enriched in tumors. Of these, C1 showed preferential overexpression of HLA-DR compared to C5 and C7 (Fig. 4a, b). Additional flow cytometry immunophenotyping showed that HLA-DR+ Tregs infiltrating BC preferentially coexpress CCR8, ICOS and IRF4 (Fig. 4e), thereby revealing immunophenotypic overlap with activated Tregs described in other tumors. CCR8hi ICOShi Tregs comprised about 40% of all BC-infiltrating Tregs (Supplementary Fig. 4d). These data indicate that a highly suppressive Treg phenotype can also be found in BC[35].

We next tested whether the abundance of the activated Tregs could influence the prognostic benefit of CD127− CD39hi Trm cell infiltration in BC. To this aim, we further stratified BC patients from the METABRIC dataset based on combinations of enrichment of CD127− CD39hi Trm and CCR8hi ICOShi (HLA-DRhi) effector Treg gene signature, the latter previously identified in NSCLC[35], and investigated OS in these subgroups (Fig. 4f, g). In this context, the presence of CD127− CD39hi Trm cells correlated with improved OS especially when CCR8hi ICOShi effector Tregs were less abundant (Fig. 4f). Among all groups, BC patients showing low levels of CD127− CD39hi Trm cells and high levels of CCR8hi ICOShi effector Tregs had the worst OS. Similar results were obtained when taking into account only luminal-like BCs (Fig. 4g), or when analyzing OS data and BC-specific survival (Supplementary Fig. 4e). Overall, these data suggest the nonredundant role of the T-cell immune microenvironment also in endocrine-dependent, luminal-like BCs.

## Discussion

In this study, we dissected the BC T-cell immune milieu by single-cell technologies and focused on CD8+ TILs to gain deep insight on tumor-infiltrating Trm subpopulations, their relationship with the immune-suppressive microenvironment, and their capability of predicting and refining BC prognosis. By analyzing numerous primary tumors, adjacent normal breast tissues, and peripheral blood, we could distinguish different CD4+ and CD8+ T-cell subsets. In our cohort, improved survival is specifically associated with a combination of subsets of CD127− CD39hi CD8+ Trm cells and CCR8hi ICOShi activated Tregs that are preferentially abundant in the tumor compared to the normal breast tissue and the peripheral blood from the same patient. Of note, the landscape of these T-cell phenotypes was shared among luminal A, luminal B, and TNBC subtypes, pointing at the relevance of the immune infiltrate also in luminal-like BCs for which immunotherapeutic strategies have been poorly investigated to date. Given the high prevalence (70–80%) of luminal-like subtypes among all surgically treated BCs every year, and their indolent nature, with possible recurrences over 20 years after surgery[36], our data are of broad applicability and prompt to investigate immunotherapeutic strategies in these tumors further.

Our results uncovered heterogeneity within the intratumoral CD8+ Trm compartment, distinguishing different subsets by combining CD127 and CD39 expression. The latter recently proposed to identify those T cells specific for tumor-derived epitopes among melanoma, colon, ovarian, and lung TILs[15,16], was mostly confined to CD127− Trm cells, which also expressed significantly higher levels of HLA-DR, GZMB, and the checkpoint molecules PD-1 and, at a lesser extent, TIGIT, and stained more frequently for the production of CD107a compared to CD127+ CD39lo Trm cells upon nonspecific stimulation ex vivo. Although both expression of CD127 and increased IL-2 production upon ex vivo stimulation may suggest early differentiation among Trm cells, the two subsets did not differ in the expression of TCF7, a master transcription factor regulating the persistence and the stem-like potential of peripheral and tumor-infiltrating precursors of exhausted CD8+ T cells (Tpex)[37], thereby suggesting a poor overlap between CD127+ CD39lo Trm and Tpex. A third subset of CD127− CD39lo Trm appeared intermediate between the two additional populations of Trm on the basis of their immunophenotypic and functional profiles. Their specific role in BC will require more in depth studies.

A limitation of this study is the missing direct demonstration of tumor reactivity by the different CD8+ Trm cell subsets. Nevertheless, the association of CD127− CD39hi, but not of CD127+ CD39lo Trm cells with improved OS and BC-specific survival in the METABRIC cohort, their increase presence at the tumor site compared to adjacent tissue and the peripheral blood, and in metastatic vs. nonmetastatic lymph nodes suggests that CD39 is a marker of bona fide tumor-reactive T cells also in BC. In line with this conclusion, a study recently published while this paper was under review mechanistically linked the infiltration of CD39hi PD-1hi T cells to metastatic dormancy in BC[38]. Collectively, these results thus extend a previous study in BC patients on the prognostic role of CD8+ CD103+ Trm cells as a whole[12] and pinpoint those T cells that are preferentially associated with delayed tumor growth with enhanced precision. Notably, the presence of CD127− CD39hi Trm cells in luminal-like BC seems relatively low compared to other tumors, such as NSCLC, head and neck squamous cell carcinoma, and colorectal cancer[15,16], thereby requiring strategies capable to favor the infiltration or proliferation in situ of these cells to enhance tumor control.

In tumors, the abundance of CD39hi Trm cells directly correlated with that of Tregs, which instead suppress antitumor immunity. This finding may appear counterintuitive, but a possible explanation could be that enhanced immunosuppression is put in place when the immune system is more activated

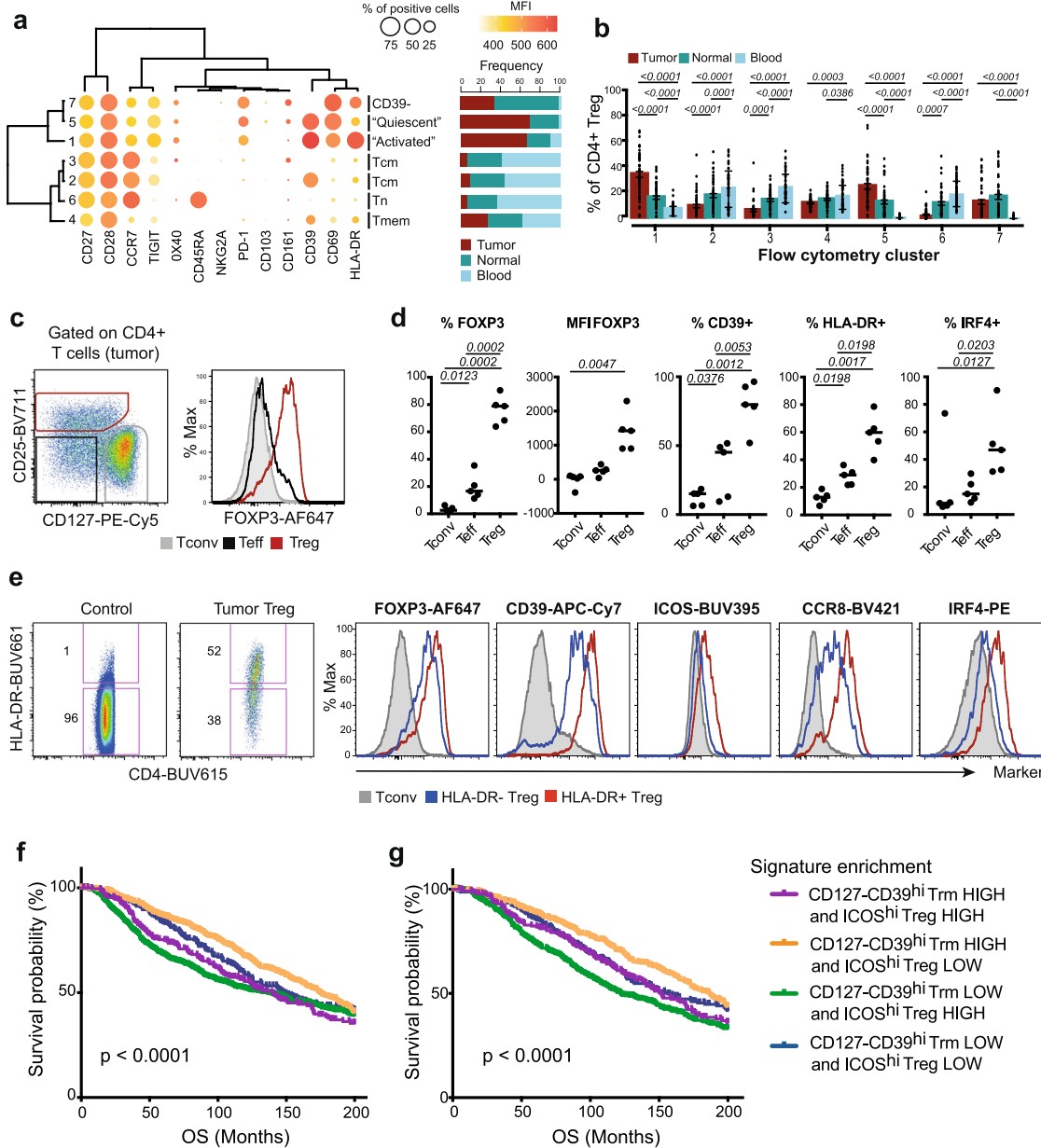

**Fig. 4 Abundance of CCR8$^{hi}$ ICOS$^{hi}$ IRF4+ effector Tregs hinders the prognostic benefit of Trm cells. a** CD127− CD25+ CD4+ Tregs were isolated by manual gating ($n = 54$ biologically independent samples), concatenated, and analyzed by PhenoGraph. Data are displayed as in Fig. 1b. **b** Bar plots showing the frequency of Treg PhenoGraph clusters, depicted as mean ± SEM, in peripheral blood, normal breast tissue, and tumor samples ($n = 54$ biologically independent experiments). Numbers indicate the exact $p$ value for tumor vs. peripheral blood or normal tissue samples; two-way ANOVA with Bonferroni post hoc test. **c** Representative flow cytometry plot identifying Treg, conventional (Tconv) and effector (Teff) T cells on the basis of CD25 and CD127 expression combinations, and FOXP3 expression in these subsets from a tumor sample. **d** Median summary of FOXP3 expression (percentage and MFI) and expression of CD39, HLA-DR, and IRF4 by Tregs, Tconv, and Teff cells gated as in **c** ($n = 5$ biologically independent samples). Numbers indicate the exact $p$ value; one-way ANOVA. **e** Representative flow cytometry plot showing the identification of HLA-DR+ and HLA-DR− Tregs in tumor (Tconv from the peripheral blood of a healthy control is shown as control staining) and flow cytometry histograms showing the expression of defined markers by HLA-DR+ and HLA-DR− Tregs and Tconv infiltrating a tumor. Similar data were obtained from four more patients. Kaplan–Meier OS curves in the METABRIC dataset in the global population (**f**; $n = 1894$) or in luminal-like BCs (**g**; $n = 1436$) according to high or low enrichment of CD127− CD39$^{hi}$ Trm or CCR8$^{hi}$ ICOS$^{hi}$ (labeled as ICOS$^{hi}$) gene signatures. The mean $z$-score value was used to classify tumor samples into LOW and HIGH expression groups. $p$ values were calculated applying the log-rank (Mantel–Cox) test.

(indeed, CD39$^{hi}$ Trm cells also express the activation marker HLA-DR+) and effective tumor rejection is occurring. Nevertheless, we have further reported that a heterogeneous population of Tregs exists also in BC, similar to other solid tumors, as recently reported by our group[35], and that the BC patient population is heterogeneous as far as the content of Trm and Treg subsets is concerned. In this regard, it was interesting to show that

the prognostic benefit of CD39$^{hi}$ Trm cells is hindered by the presence of a highly suppressive subset of CCR8$^{hi}$ ICOS$^{hi}$ IRF4+ Tregs orchestrating inhibition of antitumor immunity in the TME[35]. The importance of this association is manifold: first, measurement of multiple subpopulations of tumor-infiltrating immune cells along with their molecular status is critical to identify patients with improved prognosis, and second, strategies

aiming at promoting effector functions of TILs in luminal-like BCs, for instance checkpoint blockade as it is currently tested in clinical trials, may be hampered in a subset of patients because of the excessive infiltration of highly immunosuppressive Tregs. It is thus tempting to speculate that these individuals may require combinatorial strategies that also target Tregs for successful tumor regression. The recent observation that CDK4/6 inhibitors, now widely employed to treat luminal-like BCs, not only restrain cancer cell proliferation but also favor antitumor immunity by inhibiting CD4+ Treg proliferation in preclinical models, may be important in this regard. Clinical trials are currently ongoing to test the efficacy of checkpoint blockade in association with these agents.

In summary, our high-dimensional single-cell measurements defined a signature of T-cell subsets involving CD127− CD39$^{hi}$ Trm and CCR8$^{hi}$ ICOS$^{hi}$ IRF4+ effector Tregs, with high and low abundance, respectively, that is associated with good prognosis in the long term. These data urge to integrate multiparameter measurements of patients' specimens with a defined set of markers to be translated in clinical practice to predict disease progression and/or response to treatment in combination therapies.

## Methods

**Study design**. Experiments were approved by the Humanitas Clinical and Research Center Internal Review Board (Prot. No. Humanitas ONC-OSS-02-2017). Patients received the informed consent forms and signed them in accordance with the Declaration of Helsinki. In this study, we included patients affected by early stage BC, consecutively treated with surgery (lumpectomy or mastectomy with sentinel lymph node biopsy and, if clinically indicated, axillary dissection) at Humanitas Cancer Center Breast Surgery Unit. Tumor ($n = 54$) and breast gland normal tissue from the same surgical specimen ($n = 54$) and lymph nodes (either metastatic or tumor-free; $n = 12$ and 9, respectively) were obtained in the operating room in a sterile field by the surgeon. Blood samples (12–15 mL; $n = 54$) were collected by venipuncture before anesthesia induction. None of the patients received neoadjuvant chemotherapy or endocrine therapy. Details on patients' characteristics are summarized in Supplementary Tables 2 and 4.

**Sample collection and processing**. Blood samples were collected in Vacutainer EDTA tubes (BD). Tumor, normal tissue, and lymph node samples were collected in RPMI-1640 pure medium (Sigma-Aldrich). Peripheral blood mononuclear cells (PBMCs) were isolated from the blood using a density gradient centrifugation after blood stratification on Ficoll-Paque Premium (GE Healthcare). Freshly obtained surgical specimens were processed into single-cell suspensions by mincing and by subsequent enzymatic digestion using Liberase TL (Sigma) for 20 min at 37 °C. Samples were then passed through a 100-μm cell strainer (Falcon) and washed with physiological saline solution (NaCl 0.9%; Baxter). The cells were resuspended in a solution containing 10% dimethyl sulfoxide in fetal bovine serum (FBS), then frozen in liquid nitrogen according to standard procedures.

**High-dimensional single-cell analysis by flow cytometry**. Samples were analyzed by flow cytometry after staining with fluorochrome-conjugated mAbs listed in Supplementary Table 1, as previously described[39]. For ex vivo immunophenotyping experiments, frozen samples were thawed in RPMI-1640 medium supplemented with 10% FBS (Sigma-Aldrich), 1% penicillin–streptomycin, and 1% ultraglutamine (both from Lonza) (hereafter referred to as R10), supplemented with 20 μg/mL DNase I from bovine pancreas (Sigma-Aldrich). After extensive washing with PBS (Sigma-Aldrich), the cells were stained immediately with the combination of mAbs listed in Supplementary Table 1, together with Zombie Aqua Fixable Viability kit (Biolegend). All mAbs were previously titrated on human PBMCs and used at the concentration giving the best signal-to-noise ratio, as described[40]. Both surface markers and chemokine receptors were stained for 20 min at room temperature. All data were acquired on a FACS Symphony A5 flow cytometer (BD Biosciences) equipped with five lasers (UV, 350 nm; violet, 405 nm; blue, 488 nm; yellow/green, 561 nm; red, 640 nm; all tuned at 100 mW, except UV tuned at 60 mW) and capable to detect 30 parameters. Flow cytometry data were compensated in FlowJo (FlowJo LLC) by using single stained controls (BD Compbeads incubated with fluorescently conjugated antibodies)[40].

**Computational analysis of flow cytometry data**. Flow Cytometry Standard (FCS) 3.0 files were imported into FlowJo software version 9, and analyzed by standard gating to remove aggregates and dead cells, and identify CD4+ and CD8+ T cells. A total of 1000 of each CD8+ and CD4+ T cells per sample were subsequently imported in FlowJo version 10. Data were then biexponentially transformed and exported for further analysis in Python version 3.7.3 using a custom-made script

(available at https://github.com/luglilab) that makes use of PhenoGraph from the scikit-learn package. Tissue samples were labeled with a unique computational barcode, converted into comma separated (CSV) files, and concatenated in a single matrix using the "concat" function in pandas (https://pandas.pydata.org/). The $K$ value, indicating the number of nearest neighbors identified in the first iteration of the algorithm, was set to 1000 and 1500 to generate data in Figs. 1b and 4a, respectively. UMAP applications were run in Python and visualized using FlowJo software version 10. Clusters representing <1% were excluded from subsequent analyses. The data were then reorganized and saved as new FCS files, one for each cluster, that were further analyzed in FlowJo to determine the frequency of positive cells for each marker and the corresponding MFI, subsequently visualized by the use of ballon plots. Metaclustering of these data was performed using the gplots R package. Hierarchical metaclustering of all samples, based on the frequency of PhenoGraph clusters, was performed in R based on the Euclidean distance and Ward linkage. Pearson's correlation analysis was used to investigate the relationship between CD4+ and CD8+ clusters.

Hierarchical metaclustering of all samples, based on the frequency of PhenoGraph clusters, was performed in R based on the Euclidean distance and Ward linkage.

**scRNA-seq data processing and in silico sorting of T-cell subsets**. Unnormalized scRNA-seq counts from eight BCs were downloaded from Gene Expression Omnibus dataset (GSE114725). Analysis was restricted to the cells from tumoral samples ($n = 21,253$) and with a $CD3E$ gene expression level >1.87 ($n = 3637$) based on the normalized expression levels ($E$). Subsequently, $CD3E+$ cells were separated into CD4+ ($n = 475$) and CD8+ ($n = 1167$) based on the normalized expression levels of $CD4$ ($E > 1.56$) and $CD8A$ ($E > 2.07$). Thresholds of expression were determined by calculating 25th percentiles of the mRNA distribution, by excluding cells of the lower percentile and cells double positive for $CD4$ and $CD8A$. Data were imported into R version 3.5.1 and analyzed with Seurat version 3.0.1[41]. Genes detected in fewer than three cells or cells containing fewer than 200 features were excluded from the analysis. The resulting datasets were normalized and log-transformed using the "ScaleData" function in Seurat. Cluster analysis was performed by using the "FindClusters" function with a resolution of 1.2 and 1 for CD4+ and CD8+, respectively. Clusters with ambiguous or unknown gene expression (i.e., expressing genes not related to T cells) were removed both from the UMAP plot and from the differential analysis. DEGs were identified among clusters through the "FindAllMarkers" function in Seurat.

**Analysis of effector molecule production**. To induce cytokine production, cells were plated in U-bottom 96-well plate and stimulated with PMA (10 ng/mL; Sigma-Aldrich) and Ionomycin (1 μg/mL; Sigma-Aldrich) or left unstimulated at 37 °C, in the presence of Golgi Plug (brefeldin A, 1 μg/mL; BD Biosciences) and Golgi Stop (monensin, 0.67 μg/mL; BD Biosciences) for 3 h. Subsequently, cells were collected, washed with PBS, and fixed and permeabilized by using the Foxp3 Transcription Factor Staining Buffer Set (eBioscience), followed by staining with mAbs listed in Supplementary Table 1. As PMA/ionomycin can randomly induce immunophenotypic changes in cells from some, but not all the individuals, thereby compromising Trm subsets identification, we restricted analysis in Fig. 2e, f to samples that maintained the original architecture compared to the unstimulated control. In Fig. 2f, the percentage of cytokine-producing cells is referred to the stimulated condition subtracted of its background in the unstimulated condition. For CD107a staining, one sample was excluded from the statistical analysis due to faulty antibody staining.

**Total RNA-sequencing analysis**. CD127+ CD39$^{lo}$ and CD127− CD39$^{hi}$ cells, pregated as Aqua− CD8+ CD69+ CD103+ (Supplementary Fig. 3a), were purified from thawed tumor samples by using a FACSAria cell sorter (BD Biosciences). A list of antibodies used for FACS sorting is shown in Supplementary Table 2. Patient characteristics of the samples used for FACS sorting are listed in Supplementary Table 5. Cells were sorted in 10 μL PBS 1X (Sigma-Aldrich), spun down immediately, and stored frozen at −80 °C in a maximum volume of 7 μL PBS 1X (Sigma-Aldrich). Library preparation was performed by starting directly from 500 cells per sample using the SMART-Seq Stranded Kit (Clontech-Takara) and following the manufacturer's protocol. Libraries were qualitatively assessed by using TapeStation 4200, quantified by Qubit Fluorometer, then multiplexed in equimolar pools, and sequenced on a NextSeq-550 Illumina Platform, generating on average 32.9 million 75bp-PE reads per sample. The sort procedure was performed on six different luminal-like BC samples and, based on the number of cells obtained after sort of each population, four samples were deemed suitable for bulk total-RNA-seq. After quality check, three replicates of CD127− CD39$^{hi}$ and four replicates of CD127+ CD39$^{lo}$ Trm cells were deemed suitable for differential analysis. Paired-end strand-specific reads were then aligned to the human genome (GENCODE Human Release 32; Reference genome sequence: GRCh38/hg38) using STAR (version 2.5.1b)[42]. Alignments were performed using default parameters. Reads associated with annotated genes were counted with the STAR aligner option -quantMode geneCounts. Differential gene expression between human Trm cell subsets was assessed using the edgeR package (version 3.22.5)[43]. Benjamini–Hochberg correction was applied to estimate the FDR.

**Overrepresentation analysis (RNA-seq).** GSEA was performed using GSEA software (version 3.0; Broad Institute, MIT) and gene list ranked based on $log_2$ fold changes. The GSEA was conducted in preranked mode with scoring scheme "classic" and 1000 permutations. The maximum gene set size was fixed at 500 genes, and the minimum size fixed at 15 genes. The gene signature was retrieved from the Molecular Signatures Database (MSigDB v7.1).

**Survival analysis.** Transcriptomic and clinical data related to the Molecular Taxonomy of Breast Cancer International Consortium[34] dataset were downloaded from the cBioPortal platform (http://www.cbioportal.org)[44]. For the three signatures (CD127− CD39hi Trm; CD127+ CD39lo Trm; CCR8hi ICOShi IRF4+ effector Treg), normalized signal values were converted into $z$-scores using the formula: $z$-score $= (X − \text{average}(X))/\text{stdev}(X)$, where $X$ corresponds to the expression values. The average and standard deviation were calculated from the expression of $X$ across all samples considered for this study. The mean $z$-score value was used to classify tumor samples into LOW and HIGH expression groups. Survival analysis was performed with GraphPad Prism using the Kaplan–Meier approach and applying the log-rank (Mantel–Cox) test to estimate survival curves comparison.

**Statistics and reproducibility.** Statistical analyses were performed using Prism version 7.0c (GraphPad) or R software version 3.4.4. Significance was assigned at $p < 0.05$, unless stated otherwise. Nonsignificant differences are not depicted for simplicity. Specific tests are indicated in the relevant figure legends. Statistical analysis was performed from at least three independent experiments, where applicable, and the number of experiments is indicated in the figure legends. Otherwise, independent biological replicates were included in a single experiment. Justifications for data exclusion are present in the "Methods" section. The sample size was chosen based on previous experience.

**Reporting summary.** Further information on research design is available in the Nature Research Reporting Summary linked to this article.

## Data availability

The bulk RNA-seq data reported in this paper have been deposited in the Gene Expression Omnibus at NCBI (https://0-www-ncbi-nlm-nih-gov.brum.beds.ac.uk/geo/) under the accession code GSE154842. Unnormalized single-cell RNA sequencing counts from eight breast cancers were downloaded from Gene Expression Omnibus dataset (GSE114725). Transcriptomic and clinical data related to the Molecular Taxonomy of Breast Cancer International Consortium[34] dataset were downloaded from the cBioPortal platform (http://www.cbioportal.org)[44]. Bulk RNA-seq data of Tregs were downloaded from Gene Expression Omnibus dataset (GSE128822)[35]. Source data are provided as Supplementary Data.

## Code availability

Scripts to analyze the flow cytometry data and RNA-seq data are available upon reasonable request.

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

## Acknowledgements

The authors thank the other members of the Laboratory of Translational Immunology for critical discussions. This work was funded by Intramural Research Funding of the Humanitas Clinical and Research Center (to E.L. and A.L.) and, in part, by the Associazione Italiana per la Ricerca sul Cancro (IG 20676). J.B. was supported by a fellowship from the Fondazione Italiana per la Ricerca sul Cancro-Associazione Italiana per la Ricerca sul Cancro. K.P. was supported from a 2020 fellowship from the Fondazione Umberto Veronesi. The purchase of a FACSymphony A5 was defrayed in part by a grant from the Italian Ministry of Health (agreement 82/2015).

## Author contributions

A.L., C.S. and E.L. conceived the study; A.L., C.S., G.A., J.B., K.P., F.S.C., C.P. and E.M.C.M. performed experiments; A.L., C.S., G.A., E.M. C.M. and E.L. analyzed data; A.L., V.E., L.D.T., B.F., A.T., C.T., M.R., and A.S. provided clinical expertise and sample collection; A.L. and E.L. supervised the study; A.L., C.S. and E.L. wrote the manuscript. All authors contributed intellectually and approved the manuscript.

## Competing interests

The Laboratory of Translational Immunology receives reagents in kind as part of a collaborative research agreement with BD Biosciences (Italy). A.S. has received honoraria as advisory board member from Bristol-Myers Squibb, Servier, Gilead, Pfizer, Eisai, Bayer, and Merck Sharp & Dhome; as speaker's bureau member from Takeda, Roche, Abbvie, Amgen, Celgene, Astrazeneca, Lilly, Sandoz, Novartis, Bristol-Myers Squibb, Servier, Gilead, Pfizer, Arqule, and Eisai; and for consultancy from Arqule. All other authors declare no competing interests.
