## [Peer Review File · Communications Biology]

Reviewers' Comments:

Reviewer #1:

Remarks to the Author:

This study presents an elegant and comprehensive single-cell analysis of T cells infiltrating human breast cancers. Two subsets of resident memory T cells are identified and characterised in their phenotype, functions and gene profile; only one of the two correlates with improved survival, and this becomes especially true in association with low frequencies of ICOS+ Tregs. The study is conducted with a combination of advanced flow cytometry and bioinformatics analyses, and is clearly and accurately presented. It is especially relevant in the human tumor immunology field, since a significant number of primary tumours have been analysed at high resolution, and significant association with survival has been proved for the key parameters.

My detailed comments are as follows:

- In figure 1B and 4A, I cannot see colours in the legends showing the iMFI scale values.
- In figure 2B, I suggest using the same scaling intervals in the y axis throughout all plots (0-50-100 versus 0-20-40-60-80-100)
- In figure 1C, the red lines indicating positive cells should be extended to the right in the case of TNF.
- The sentence at line 209 "Thus, BC tumours preferentially harbour a subset of CD127- HLA- DR+ CD39hi Trm cells that is more functional than CD127+ HLA-DR- CD39lo Trm cells" is not clear: from figure 1B, it seems that these two subsets (roughly corresponding to clusters 7 and 6/8 of CD8) are both expanded and similarly represented in tumours compared to normal tissue and blood. This statement needs to be confirmed through more appropriate analyses.
- Figure 2E is mentioned in the text after figure 2C and D and should be moved up.
- In Suppl. Figure 2, the biexponential scaling for the y axes is slightly different (but the gates are equal) between metastatic and non-metastatic lymph node: please check.
- Tregs correlate positively with CD127- CD39 hi Trm (suppl. Fig. 4), which is somehow counterintuitive since the former is considered detrimental and the latter protective. The survival data shown in figure 4 speak in favour of this view. Can the authors propose an explanation for this finding?
- The CD127+ Trm population described here resemble memory-like exhausted effectors that have been observed in viral chronic infections (see for instance 10.1038/ncomms15050) and characterised by Tcf1 expression. Is Tcf7 gene differentially expressed in the two Trm subsets? Some hypotheses about the nature of these two Trm subsets should be elaborated in the discussion.

Reviewer #2:

None

Reviewer #3:

Remarks to the Author:

Remarks to the authors:

In the manuscript entitled "Single-cell profiling defines the prognostic benefit of tumor-reactive tissue resident memory T cells in luminal-like breast cancer", the authors investigated the composition of the tumor immune cell infiltrate in luminal-like breast tumors, focusing their analysis on CD8 T cells. Using a high-dimensional flow cytometry approach, the authors found that cells with a tissue-resident memory (Trm) CD8 phenotype were phenotypically and functionally heterogeneous in those tumors. Among the Trm CD8 T cells, a subset of CD127-CD39hi cells was preferentially present in the tumor compared to the adjacent normal breast tissue and peripheral blood. Patients whose tumors were enriched for a gene signature specific for CD127- CD39hi Trm CD8 T cells had a better prognosis. Nevertheless, the positive prognostic benefit was lost in the presence of a highly suppressive Treg cell population.

As mentioned by the authors, it is true that most studies on breast cancer focus on characterizing

the tumor T cell infiltrate in TNBC (apart from some data presented in the paper from Savas P. et al, Nat Med 2018). It is thus interesting to see an analysis of the T cell infiltrate in luminal-like breast cancer, which is the most common breast cancer with poor responsiveness to immunotherapy.

Title:

The authors do not show any data supporting the tumor-reactivity of the CD8 T cell population they describe. Thus, the title of the manuscript should be modified.

Fig.1

Fig.1 A: It is an interesting approach to use previously published single-cell RNAseq datasets. However, the dataset they used contained a very low number of cells (3,637 cells, both CD4 and CD8 T cells from 8 patients) which might question its relevance.

Fig.1 B represents a summary of the flow cytometry analysis of 54 treatment naïve BC patients comparing peripheral blood, normal breast tissue and tumor, presented as heatmaps. I appreciate that the authors analyzed a large number of samples for that study as it helps overcome the inherent heterogeneity between patients and increase its biological significance. Unfortunately, those heatmaps are not easy to read and it would be more appropriate to present the data as t-SNE or UMAP plots, highlighting the different clusters on the plots. It would also help understand the proportion of each cluster among the total population of CD4 or CD8 T cells.

CD4 panel: I have some concerns regarding the NKG2A staining on these cells. Data in the literature clearly indicate that NKG2A is expressed almost exclusively by CD8 cells in the T cell compartment (Sheu BC et al., Cancer Res 2005; van Montfoort N et al., Cell 2018). I would appreciate if the authors could provide flow plots of NKG2A expression on CD4 and CD8 T cells on 5 to 10 patients to determine if there is indeed expression of this receptor on CD4 T cells. On another note, it is unfortunate that FOXP3 was not included in the panel as the identification of Treg cells using CD127 and CD25 is not very distinct in the tumor compared to peripheral blood due to the high degree of cellular activation at the tumor site. To help the reader appreciate the quality of the staining, it might be necessary to present actual flow data of the gating strategy from one or two representative patients, including the three different tissue sources.

CD8 panel: It is surprising to see OX40 expression on CD8 T cells ex vivo. Data from several labs, including ours, has clearly shown that OX40 is present at low levels on ex vivo CD8 T cells. The authors should provide representative data to support OX40 expression on CD8 T cells in BC tumors (use CD4 T cells or Treg cells as a comparison).

Fig.2:

On line 187, the notion that CD39, CD127 and HLA-DR expression was mutually exclusive in CD8+CD103+ is not supported by data presented in Fig.2B. If it were the case, CD127+ Trm would be negative for HLA-DR and CD39, which they are not. Presenting data as MFI might be more appropriate. Similarly, many cells can be positive for PD-1, but express it at varying levels (and different MFI can have a different biological meaning).

Why did the authors focus on CD127+ and CD127- cells in the Trm? There is no clear justification in the text.

The authors indicate that similar phenotypes of CD127+ and CD127- Trm in the CD4+ and CD8+ compartment are shown in Suppl. Fig.2. However, this figure only shows data for CD8 T cells. The authors should add that data or modify the sentence. The flow plots for non-metastatic LN presented in Suppl Fig.2 are no representative of the data collected by the authors (see Fig.2 E). Indeed, there is a significant proportion of CD8 Trm that express CD39hi also in non-metastatic LN. Those plots should be replaced by plots from a more representative patient. CD39 is usually expressed by T cells chronically stimulated via their TCR. Do the authors have an explanation as to why so many CD39hi are found in non-metastatic LN? Is there a difference in MFI for CD39 between metastatic and non-metastatic LN? It would be very useful to the reader to have a summary of the frequency of the CD127+CD39lo and CD127-CD39hi cells in the CD8 Trm in the blood, normal tissue and tumor from BC patients.

On line 198, the authors conclude that their data corroborate the evidence that CD39 identifies tumor-reactive CD8+ Trm cells in BC. However, from their data it is not possible to conclude that CD39 identifies tumor-reactive CD8 Trm cells in BC. Additional experiments such as analysis of the TCR repertoire and tumor-reactivity experiments would be necessary to prove that statement. In Fig.2C, the authors use HLA-DR to distinguish CD127+ and CD127- cell subsets. In order to do

that, they need to show correlation between those two markers and also between HLA-DR and CD39 on the CD8 T cells in the patient samples they are analyzing. The gating for positive cells presented in Fig.2C is not correct. There are clearly two populations of cells for CD107a, TNF and IFN-g. This separation between the negative and the positive population should be used to distinguish positive from negative cells as it takes into consideration the autofluorescence of the cells.

Fig. 3:

In Suppl Fig.3, the authors show their cell sorting gating strategy to purify cells for RNAseq analysis. I am confused as to where they placed the gate for the CD39^{lo} population. They are missing many cells that are CD39^{lo} and CD127⁺. Is this a mistake during the figure preparation for that particular plot or were the cells sorted this way for the study?

More information regarding the bulk RNAseq dataset would be welcome. How do the samples distribute when using a classic PCA approach? Do they group by subsets or by patients? A volcano plot would help better show FC and adjusted p-values for the DEG.

The GSEA analysis is very confusing. The authors claim that the CD127-CD39^{hi} CD8 T_{rm} cell population was enriched in signatures related to stemness whereas the CD127⁺CD39^{lo} CD8 T_{rm} were characterized in gene signatures related to T cell exhaustion. Many articles in the literature to date have shown the opposite - CD39 as a marker of T cell exhaustion (Canale FP, Cancer Res 2018; Duhon T, Nat Comm 2018; Gupta PK, PLoS Pathog 2015). This makes me wonder if the CD127-CD39^{hi} and CD127⁺CD39^{lo} dataset have been inverted.

The list of genes used for the signature needs to be indicated in the publication. How distinct is their signature compared to the one from Savas et al., Nat Med 2018? How does their gene signature compare to using CD8a or itgae expression to predict BC patient's survival (as shown by Ganesan et al in patients with lung cancer - Ganesan et al, Nat Immunol 2017)? How does it compare to the signature from Savas et al?

Fig. 4:

I agree that it is very important to study the interplay between the different immune cell populations in the tumor microenvironment (TME). However, in order to accurately and confidently analyze Treg cells in the tumor, it is highly recommended to use an anti-FOXP3 Ab in the flow panel, especially in the absence of functional downstream experiments.

Fig.4A: The Pearson correlation analysis revealed that CD127-CD39^{hi} CD8 T cells were directly correlated with cells in CD4⁺ cluster 5 which they claim contain Treg cells. For that reason, the authors decided to look at Treg cells in the TME, more precisely at the two different subsets of Treg cells they recently identified based on IRF-4 expression (Alvisi G et al., J Clin Invest 2020) which is interesting. In that case, Fig.4A does not bring any useful information apart from suggesting that not all the cells in cluster 5 are Treg cells. It would be more appropriate to directly look at Treg cells in the tumor and determine if those two cell populations are also found in the TIL of BC patients.

Based on the literature, Treg cells are believed to be negative for NKG2A and CD103. The authors should show representative flow plots demonstrating NKG2A and CD103 expression on Treg cells in their dataset (mouse Treg cells express CD103).

Fig.4B: the gate drawn to identify Treg cells shown in this figure is not appropriate. As drawn, it might include a significant number of non-Treg CD4 T cells which could bias the downstream analysis. A tighter gate around the CD127^{lo}CD25^{hi} cells would be better. The authors should present a summary of the frequency of the markers IRF-4, CD39 and HLA-DR on Treg cells in their dataset.

Fig.4C and D: the authors show that in the presence of a strong ICOS^{hi}CCR8^{hi} Treg signature, the positive correlation of CD127-CD39^{hi} CD8 T_{rm} with survival is reduced in BC patients. Based on this analysis, do the authors believe that both subsets interact in the tumor? If so, are they planning to analyze the spatial location of those two subsets by IHC?

In conclusion, I think the study by Losurdo et al is interesting, in particular the association of CD127-CD39^{hi} CD8 T_{rm} gene signature with better survival in patients with luminal-like BC and the loss of this survival advantage in the presence of a strong signature for ICOS^{hi}CCR8^{hi} Treg cells. However, it still remains to be seen how this signature performs in comparison to the signature previously published by Savas et al or using CD8a gene and/or itgae genes to predict survival. Also, a more in-depth analysis of the CD8 T cell compartment and Treg cell compartment

would be welcome. For the latter, inclusion of a FOXP3 antibody to the panel would greatly reduce the risk to include non-Treg cells in the analysis. On another note, the claim that CD127-CD39hi CD8 Trm are tumor-reactive is based only on previous publications that showed that such cells are enriched in tumor-reactive T cells in HNSCC, CRC and melanoma. It would be useful to confirm that in BC as well. Finally, this analysis shows that the composition of the T cell infiltrate in luminal-like BC does not appear to be significantly different from TNBC, suggesting that their poor responsiveness to immunotherapy might be due to other characteristics specific for that cancer-type.

Point-by-point response to the Reviewers

The authors would like to thank the Reviewers for the valuable comments. We think that the manuscript has greatly improved in its revised form. We hope our changes are acceptable.

Reviewer 1

1. In figure 1B and 4A, I cannot see colours in the legends showing the iMFI scale values.

Author: we apologize for this inconvenience. We have noticed that opening the document with Acrobat Reader obviates the problem. In any case, this is not a problem in the new visualization approach we are proposing in the revised version (see below).

2. In figure 2B, I suggest using the same scaling intervals in the y axis throughout all plots (0-50-100 versus 0-20-40-60-80-100)

Author: we modified **Fig. 2B** as requested

3. In figure 1C, the red lines indicating positive cells should be extended to the right in the case of TNF.

Author: to also respond to Reviewer 3's request and simplify visualization of the data, we modified **Fig. 2B** where flow cytometry histograms were replaced by dot plots showing the production of effector molecules by subsets of HLA-DR- and HLA-DR+ Trm cells. Gates identifying positive cells were placed on the basis of the unstimulated sample.

4. The sentence at line 209 "Thus, BC tumours preferentially harbour a subset of CD127- HLA- DR+ CD39hi Trm cells that is more functional than CD127+ HLA-DR- CD39lo Trm cells" is not clear: from figure 1B, it seems that these two subsets (roughly corresponding to clusters 7 and 6/8 of CD8) are both expanded and similarly represented in tumours compared to normal tissue and blood. This statement needs to be confirmed through more appropriate analyses.

Author: We apologize if our statement was not clear. We modified the sentence (now at line 218) in order to better explain the enrichment of CD127- HLA- DR+ CD39hi Trm cells in BC tumors compared to adjacent tissue and the blood. The Reviewer reports that the abundance of clusters 7 and 6/8 is not different between the tumor and the blood or the adjacent tissue. ANOVA analyses included in the first submission (now in **Fig. 1C**) revealed the increased presence of CD8+ Trm clusters 6, 7 and 8 in tumors compared to PB. This increase was also noted in comparison to the adjacent tissue as far as cluster 7 was concerned (please note that we revised figure 1 to address Reviewer 3's request, and now include a new bar graph in **Fig. 1C** depicting the frequencies of clusters in the different tissues, previously displayed with the use of balloon plots). To strengthen our conclusions, and also in response to Reviewer 3, we quantified subsets of CD127- CD39hi and CD127+ CD39lo subsets of CD8+ Trm cells by manual gating, and found dynamics similar to those identified by Phenograph analysis (depicted in a new **Supplementary Fig. 2B**).

5. Figure 2E is mentioned in the text after figure 2C and D and should be moved up.

Author: We thank the reviewer, we modified **Fig. 2** accordingly

6. In Suppl. Figure 2, the biexponential scaling for the y axes is slightly different (but the gates are equal) between metastatic and non-metastatic lymph node: please check.

Author: we apologize for the inconvenience and modified **Supplementary Fig. 2C** accordingly.

7. Tregs correlate positively with CD127- CD39^{hi} Trm (suppl. Fig. 4), which is somehow counterintuitive since the former is considered detrimental and the latter protective. The survival data shown in figure 4 speak in favour of this view. Can the authors propose an explanation for this finding?

Author: when taking into account the correlation between all different CD4+ and CD8+ clusters we could indeed find a positive, significant correlation between CD4+ Tregs and CD127- CD39^{hi} Trm. We agree that this is counterintuitive on the basis of the Reviewer's interpretation. However, our observation performed in the bulk cohort of patients does not necessarily contrast with the survival results presented in Figure 4 where we hypothesized that the cohort is heterogeneous, and thus subdivided patients according to the different levels of CD39^{hi} Trm and ICOS^{hi} CCR8^{hi} Tregs. A second, alternative interpretation of the data is that more immunosuppression is required when the tumor is infiltrated by a high number of tumor-reactive CD39^{hi} cells that are also more activated compared to bystander cells (indeed, CD39^{hi} cells are also HLA-DR+). We modified the Discussion to include these aspects.

8. The CD127+ Trm population described here resemble memory-like exhausted effectors that have been observed in viral chronic infections (see for instance 10.1038/ncomms15050) and characterised by Tcf1 expression. Is Tcf7 gene differentially expressed in the two Trm subsets? Some hypotheses about the nature of these two Trm subsets should be elaborated in the discussion.

Author: along with other groups, we have previously characterized precursors of exhausted T cells expressing TCF-1 and CXCR5 in human tumors (Tpex; see Brummelman et al., J Exp Med, 2018). These cells also express additional memory markers such as CD27 and CD28 that instead are not expressed by CD127+ CD39^{lo} Trm cells reported in the present study. In our bulk RNAseq dataset, we investigated the expression of TCF7 and found that it is not differentially expressed between CD127+ CD39^{lo} compared to CD127- CD39^{hi} Trm cells (FDR value = 0.058; for this reason not appearing in Table S3). On the basis of these data, we conclude that TCF-1+ Tpex and CD127+ CD39^{lo} Trm cells are different populations of CD8+.

Reviewer 3

1. Fig.1 A: It is an interesting approach to use previously published single-cell RNAseq datasets. However, the dataset they used contained a very low number of cells (3,637 cells, both CD4 and CD8 T cells from 8 patients) which might question its relevance.

Author: We agree with the Reviewer that the original dataset did not contain a large number of T cells. This could be a concern for the identification of rare subsets but not for our main purpose which was to employ a comprehensive, unbiased approach to guide the selection of markers to be included in a high-dimensional single cell flow cytometry panel. We followed this approach on the basis of recent data from our lab in non-small cell lung cancer where scRNA-seq-guided high-dimensional flow cytometry was able to identify subgroups of patients with different prognosis compared to flow cytometry panels designed on the basis of information from the literature (see Alvisi et al., J Clin Invest, 2020).

2. Fig.1 B represents a summary of the flow cytometry analysis of 54 treatment naive BC patients comparing peripheral blood, normal breast tissue and tumor, presented as heatmaps. I appreciate that the authors analyzed a large number of samples for that study as it helps overcome the inherent heterogeneity between patients and increase its biological significance. Unfortunately, those heatmaps are not easy to read and it would be more appropriate to present the data as t-SNE or UMAP plots, highlighting the different clusters on the plots. It would also help understand the proportion of each cluster among the total population of CD4 or CD8 T cells.

Author: To overcome difficulties in the identification of different clusters and their relative frequency in tissues, we show UMAPs of concatenated CD4+ and CD8+ T cells from peripheral blood, normal breast tissue and tumor samples (top rows), and color-coded UMAPs depicting clusters identified by Phenograph (bottom rows) (**Supplementary Fig. 1A**), as requested by Reviewer.

We also understand the Reviewer's concern on the flow cytometry data visualization of the integrated MFI (iMFI) approach by using heatmaps. In this case, iMFI distribution for a given antigen is normalized across samples, providing information on the relative expression across clusters (as in gene expression experiments). Thus, some antigen expression may appear high because a specific cluster of cells expresses more of that antigen compared to the rest of the clusters. As in some cases this can lead to erroneous interpretation of the results compared to more classical visualization of flow cytometry data (see for instance Reviewer's comments below on NKG2A or OX40 expression), we used a modified visualization approach that independently takes into account both the percent of antigen expression (ballon size) and MFI (color intensity) (**Fig. 1B and Fig. 4A**). Methods have been updated. We are confident that the new display can better convey our results to the readers.

3. CD4 panel: I have some concerns regarding the NKG2A staining on these cells. Data in the literature clearly indicate that NKG2A is expressed almost exclusively by CD8 cells in the T cell compartment (Sheu BC et al., Cancer Res 2005; van Montfoort N et al., Cell 2018). I would appreciate if the authors could provide flow plots of NKG2A expression on CD4 and CD8 T cells on 5 to 10 patients to determine if there is indeed expression of this receptor on CD4 T cells. On another note, it is unfortunate that FOXP3 was not included in the panel as the identification of Treg cells using CD127 and CD25 is not very distinct in the tumor compared to peripheral blood due to the high degree of cellular activation at the tumor site. To help the reader appreciate the

quality of the staining, it might be necessary to present actual flow data of the gating strategy from one or two representative patients, including the three different tissue sources.

CD8 panel: It is surprising to see OX40 expression on CD8 T cells ex vivo. Data from several labs, including ours, has clearly shown that OX40 is present at low levels on ex vivo CD8 T cells. The authors should provide representative data to support OX40 expression on CD8 T cells in BC tumors (use CD4 T cells or Treg cells as a comparison).

Author: We agree with the reviewer regarding data on NKG2A expression. As explained above, we decided to use an improved visualization approach to depict the flow cytometry data. As pointed out by the Reviewer, NKG2A expression among CD4+ T cells is not meaningful (**Fig. 1B**, CD4), in line with Reviewer's observation. Certainly, some clusters may express more of this receptor than others (as it could be appreciated from previous visualization), however it is probably not important to highlight such a difference. Similar conclusions can be drawn for OX40 expression among CD8+ T cells. These phenotypes were not mentioned in the first version of the manuscript, thus our conclusions remain unchanged. The new visualization approach confirmed clear NKG2A expression in at least one cluster of CD8+ cells, specifically, more prominent expression in CD127+ HLA-DR- CD39^{lo} Trm (**Fig. 1B**, CD8). Thus, the new visualization does not modify our conclusions while showing flow cytometry data in more classical, intuitive way.

We certainly agree with the Reviewer that CD25 and CD127 expression may be shaped by activation in the tumor microenvironment, although it is unclear to the authors why their combination should not be as distinct as in other sites. This is a common strategy for the identification and isolation of Tregs from tumors (please see for instance Plitas et al. and De Simone et al., *Immunity*, 2016 - many other papers used the same strategy). In **Supplementary Fig. 4B**, we are showing representative CD127 and CD25 expression in CD4+ T cells at different tissue sites. The CD127- CD25+ Treg population can be clearly identified from all tissues, showing an increase at tumor sites. Please see below for the response to Reviewer's concern about FOXP3 staining.

4. Fig.2: On line 187, the notion that CD39, CD127 and HLA-DR expression was mutually exclusive in CD8+CD103+ is not supported by data presented in Fig.2B. If it were the case, CD127+ Trm would be negative for HLA-DR and CD39, which they are not. Presenting data as MFI might be more appropriate. Similarly, many cells can be positive for PD-1, but express it at varying levels (and different MFI can have a different biological meaning).

Why did the authors focus on CD127+ and CD127- cells in the Trm? There is no clear justification in the text.

Author: We understand the Reviewer's concern on the possibility that not all the CD127+ Trm lack HLA-DR or CD39 expression. Initially, in Fig. 2A, we concentrated on subsets of CD127+ and CD127- Trm cells as suggested by the clustering data in Fig. 1B. From this analysis, we noticed substantial difference in CD39 expression and thus decided to further select CD127- Trm on the basis of CD39 positivity to specifically select putative tumor-specific CD8+ Trm cells, as suggested by recent data in the literature. We apologize if this was not clear. We now clarify this in the text. In Fig. 2D, we used HLA-DR in place of CD39 because its expression tended to not change following PMA/ionomycin stimulation in vitro. Otherwise, separation of subsets by FACS sorting for subsequent PMA/ionomycin stimulation and analysis of cytokine production failed on multiple attempts due to the low number of cells recovered from tumors of BC patients. Below, we are

including raw data exemplifying analysis of CD127, HLA-DR and CD39 markers among CD8+ Trm cells, recapitulating UMAP data. We are also including these data in **Supplementary Fig. 2A**. In any case, UMAP analysis of dozens of specimens indicates that it is common to observe HLA-DR and CD39 co-expression in the absence of CD127 (**Fig. 2A**). We followed the Reviewer's suggestion to show MFI of antigen expression in subsets of CD127+ and CD127- Trm cells (presented in **Fig. 2B**). As previously shown for frequency (%) of antigen expression, we could observe differences that are highly significant. Thus, CD127+ CD39^{lo} and CD127- CD39^{hi} subsets of CD8+ Trm cells are phenotypically distinct, at least on the basis of these immunophenotypic markers. The text has been modified at page 8 to better justify the analysis of these subsets.

Editorial Figure 1. FACS plot from one representative luminal-like BC patient, showing expression of CD127, CD39 and HLA-DR on pre-gated CD8+ CD69+ CD103+ cells from a tumor sample.

5. The authors indicate that similar phenotypes of CD127+ and CD127- Trm in the CD4+ and CD8+ compartment are shown in Suppl. Fig.2. However, this figure only shows data for CD8 T cells. The authors should add that data or modify the sentence. The flow plots for non-metastatic LN presented in Suppl Fig.2 are no representative of the data collected by the authors (see Fig.2 E). Indeed, there is a significant proportion of CD8 Trm that express CD39^{hi} also in non-metastatic LN. Those plots should be replaced by plots from a more representative patient.

Author: We modified the sentence in the text based on data in **Supplementary Fig. 2B**, showing the prevalence of CD127- CD39^{hi} CD8+ Trm cells in metastatic vs. non-metastatic lymph nodes. The plots in **Supplementary Fig. 2C** were replaced, showing FACS data from a patient that is more representative of the distribution.

6. CD39 is usually expressed by T cells chronically stimulated via their TCR. Do the authors have an explanation as to why so many CD39^{hi} are found in non-metastatic LN? Is there a difference in MFI for CD39 between metastatic and non-metastatic LN?

Author: It has to be noted that we collected metastatic and non-metastatic lymph nodes from locally advanced BC patients who underwent curative axillary lymph nodes dissection for clinical node positive (cN+) BC with histologically confirmed positive sentinel lymph node biopsy. Thus, we cannot exclude a recirculation, throughout the level I axillary lymph nodes chain, of tumor-

reactive T cells from tumor tissue. In **Editorial Fig. 2**, albeit a trend can be observed, we show that no statistically significant difference was observed in CD39 MFI between metastatic and non-metastatic lymph nodes.

Editorial Figure 2: CD39 MFI in metastatic and non-metastatic lymph nodes. CD39 MFI was calculated on pre-gated CD8+ CD69+ CD103+ Trm cells. ns: non statistically significant, paired t-test.

7. It would be very useful to the reader to have a summary of the frequency of the CD127+CD39^{lo} and CD127-CD39^{hi} cells in the CD8 Trm in the blood, normal tissue and tumor from BC patients.

Author: In **Supplementary Fig. 2B**, we show the frequency of both CD127- CD39^{hi} and CD127+ CD39^{lo} CD8+ Trm in tumor, normal breast tissue and peripheral blood as obtained by manual gating of flow cytometry data. Of note, a statistically significant difference ($P < 0.001$) between tumor and normal breast was observed for the frequency of the CD127- CD39^{hi} Trm subpopulation.

8. On line 198, the authors conclude that their data corroborate the evidence that CD39 identifies tumor-reactive CD8+ Trm cells in BC. However, from their data it is not possible to conclude that CD39 identifies tumor-reactive CD8 Trm cells in BC. Additional experiments such as analysis of the TCR repertoire and tumor-reactivity experiments would be necessary to prove that statement.

Author: As the Reviewer points out, we cannot firmly conclude that CD39 identifies tumor-reactive T cells also in BC. TCR sequencing, as suggested by the Reviewer, would inform on the clonality of the CD39^{hi} vs CD39^{lo} Trm cells (we believe the Reviewer would expect to see more oligoclonal repertoire in the former compared to the latter, possibly indicating recent antigen recognition) but would not reveal whether these are tumor-specific. Tumor reactivity experiments would directly inform on this regard, however isolation of relevant T cell subsets for in vitro expansion would require a large infiltrate, which is not very common in luminal-like breast cancer. Please also note that our surgery unit is currently working at its minimum because of a second wave of the SARS-CoV-2 pandemic in Italy, especially in the Lombardy region where we reside, and it will be almost impossible to obtain fresh specimens in the next months. Our conclusions related to CD39 and tumor reactivity are based on previous reports by other labs in melanoma, colon, ovarian and lung tumors (Simoni, Nature, 2018; Duhon, Nat Comms, 2018). We toned down our statements in the manuscript regarding the putative tumor specificity of the CD39^{hi} Trm cells and addressed this matter in a revised version of the Discussion. At the same time, we propose a new title, where “tumor-reactive” is replaced by “CD39 high”.

9. In Fig.2C, the authors use HLA-DR to distinguish CD127+ and CD127- cell subsets. In order to do that, they need to show correlation between those two markers and also between HLA-DR and CD39 on the CD8 T cells in the patient samples they are analyzing. The gating for positive cells presented in Fig.2C is not correct. There are clearly two populations of cells for CD107a, TNF and IFN-g. This separation between the negative and the positive population should be used to distinguish positive from negative cells as it takes into consideration the autofluorescence of the cells.

Author: In **Editorial Fig. 2** and **Supplementary Fig. 2A**, we showed representative samples regarding the expression of CD127, HLA-DR and CD39 by subsets of Trm cells. As stated above, the UMAP representation of antigen expression from the entire cohort of individuals was meant to show co-expression of these markers. Indeed, it shows that it is common to observe HLA-DR and CD39 co-expression in the absence of CD127 (**Fig. 2A**). We thank the reviewer for the comment on Fig. 2, which was modified to better visualize differences in cytokine productions from the previously identified Trm subpopulations. Gates identifying positive cells are now placed on the basis of negative controls of cytokine production (i.e., cells cultured in the absence of stimulus). Please note that **Fig. 2E** has been modified to include data from two additional patients. We confirmed previous data and obtained more robust statistical differences.

10. In Suppl Fig.3, the authors show their cell sorting gating strategy to purify cells for RNAseq analysis. I am confused as to where they placed the gate for the CD39^{lo} population. They are missing many cells that are CD39^{lo} and CD127+. Is this a mistake during the figure preparation for that particular plot or were the cells sorted this way for the study?

Author: we apologize for the mistake. We show the correct gate used for sorting CD127+ CD39^{lo} Trm cells in **Supplementary Fig. 3**.

11. More information regarding the bulk RNAseq dataset would be welcome. How do the samples distribute when using a classic PCA approach? Do they group by subsets or by patients? A volcano plot would help better show FC and adjusted p-values for the DEG.

Author: PCA analysis (**Editorial Fig. 3**) was not included in the final manuscript because not providing additional information that was relevant for our message. Should the Reviewer think it is important to show these data, we will be happy to include them in a revised version of the manuscript. Samples separated nicely on the basis of PC1. Indeed, we could identify 183 differentially expressed genes with FDR<0.05, shown in a heatmap in **Fig. 3A**. We kindly ask the Editor and the Reviewer to keep the heatmap representation because we think it is more intuitive to the general public than a volcano plot.

Editorial Figure 3. PCA analysis of all samples used in the RNAseq experiment. Each dot represents a sample and each color represents a different Trm subpopulation.

12. The GSEA analysis is very confusing. The authors claim that the CD127-CD39^{hi} CD8 Trm cell population was enriched in signatures related to stemness whereas the CD127+CD39^{lo} CD8 Trm were characterized in gene signatures related to T cell exhaustion. Many articles in the literature to date have shown the opposite - CD39 as a marker of T cell exhaustion (Canale FP, Cancer Res 2018; Duhon T, Nat Comm 2018; Gupta PK, PLoS Pathog 2015). This makes me wonder if the CD127-CD39^{hi} and CD127+CD39^{lo} dataset have been inverted.

Author: we exclude the possibility that the samples have been inverted on the basis of the clear evidence that *IL7R*, encoding CD127, a marker used for sorting, is one of the top differentially expressed genes between CD127+ CD39^{lo} and CD127- CD39^{hi} cells. We agree with the Reviewer that previous reports demonstrated that CD39 identifies terminally exhausted CD8+ T cells in humans. It is possible that the discordance in the findings at the level of gene expression between the current study and those mentioned by the Reviewer are due to the different strategies used to isolate T(rm) cell subsets or to the disease condition. In fact, Gupta et al. compared CD39^{hi} cells with CD39^{lo} bulk cells that were isolated from the peripheral blood from chronically-infected individuals (and thus not pre-gated as Trm). Duhon et al. instead focused on T cells isolated from HNSCC and ovarian tumors. Nevertheless, most of our results are in line with data from those papers, including the high-dimensional flow cytometry phenotype, the increased inhibitory receptor expression and the prognostic relevance of CD39^{hi} cells. We believe that the Reviewer has been confused mainly by the enrichment of “G6_exhausted CD8 T cells” gene set from Sade-Feldman et al., Cell, 2019. As defining the level of exhaustion of CD127+ CD39^{lo} and CD127- CD39^{hi} cells is not a major focus of the study (and in any case we did not draw major conclusions in this regard in the manuscript), we decided to remove reference to this gene set as it might be misleading for the reader. In the text, we provide a more nuanced interpretation of the gene expression data.

13. The list of genes used for the signature needs to be indicated in the publication. How distinct is their signature compared to the one from Savas et al., Nat Med 2018? How does their gene signature compare to using CD8a or itgae expression to predict BC patient’s survival (as shown by Ganesan et al in patients with lung cancer - Ganesan et al, Nat Immunol 2017)? How does it compare to the signature from Savas et al?

Author: The gene list obtained from RNA-seq data is now provided in Table S3. As suggested by the Reviewer, we tested the relevance of single genes or signatures from Savas et al. in luminal-like BC. In **Editorial Fig. 4**, we show Kaplan-Meier curves for overall survival (OS) by *ITGAE* ($P = 0.0164$) and *CD8A* ($P = 0.1486$) expression (A and B), confirming Ganesan et al data on the potential of *ITGAE* expression in stratifying prognosis, also in luminal BC cases. Interestingly, the TNBC Trm signature by Savas et al. did not have a prognostic value in luminal-like BC cases from the METABRIC cohort at median (50%) or quartile (75% vs 25%) cut-off (C and D, respectively), thereby suggesting that a specific luminal Trm signature is required in this regard. We are not including these data in the final manuscript because results are preliminary and we believe that a dedicated study would be required to directly compare TNBC and luminal-like BC. Should the Reviewer think these data are important, we will be happy to include them in a revised version of the manuscript.

Editorial Figure 4. Kaplan-Meier OS curves for luminal-like BC cases in the METABRIC consortium data set ($n=1,436$) according to high or low expression of *ITGAE* (A), *CD8A* (B), Savas et al. Trm gene signature (C and D). In A-C, the median z-score value was used to classify tumor samples into Low and High expression groups; in D curves were generated by partitioning cases in a 25:75 split based on ranked signature expression. P values were calculated applying the Log-rank (Mantel-Cox) test.

14. Fig. 4: I agree that it is very important to study the interplay between the different immune cell populations in the tumor microenvironment (TME). However, in order to accurately and confidently analyze Treg cells in the tumor, it is highly recommended to use an anti-FOXP3 Ab in the flow panel, especially in the absence of functional downstream experiments.

Author: We thank the reviewer for the comment. In order to confirm FOXP3 positivity in our CD4+ CD127- CD25+ Tregs, we performed a FACS staining on 5 consecutively-collected luminal-like BCs. In **Editorial Fig. 5** we confirm that FOXP3 is highly expressed by CD127^{lo} CD25^{hi} Tregs, both as % positive and as MFI compared to CD127^{hi} CD25- conventional T cells (Tconv) and CD127^{lo} CD25- effector T cells (Teff). The fact that not 100% of the CD127^{lo} CD25^{hi} cells are FOXP3+ should not be a concern: this is likely due to the difficulty in having a clear-cut FOXP3 staining in human cells (in

any case, FOXP3 MFI is ~10-fold higher in Tregs compared to Tconv – please also see **Fig. 4C, D**). We are confident the Reviewer is aware that obtaining a “pure” identification of Tregs is inherently difficult in humans due to the overlap of expressed markers, including FOXP3, which can be upregulated by activated CD4⁺ T cells (Allan, Int Immunol. 2007 Apr;19(4):345-54; **Fig. 4C** and **Editorial Fig. 5**). Thus, this is potentially a common problem of all studies involving human Tregs and using the largely employed FACS sorting strategy based on CD127 and CD25 expression. Also the downstream functional experiments suggested by the Reviewer would be potentially impacted by this issue. Ideally, one could identify Tregs on the basis of additional, specific markers that are expressed in the tumor, such as CCR8, ICOS, etc., although this would lead to a substantial underestimation of the Treg population, as we have recently demonstrated in Alvisi et al., J Clin Invest, 2020. In fact, in this paper, we have shown that putative subsets of intratumoral Treg cells, initially isolated as CD127⁻ CD25⁺, show multiple Treg properties, including:

- The capability to suppress autologous CD4⁺ Tconv proliferation from the peripheral blood (Alvisi et al., **Fig. 2E**).
- gene expression overlap with that of an intratumoral Treg-specific signature obtained from a single cell RNAseq dataset from lung cancer (Alvisi et al., **Supplementary Fig. 2C**). The single cell analysis of thousands of genes simultaneously ensures the identification of a virtually “pure” Treg population that is free of possible contaminants and that can be used as a reference.
- high levels of FOXP3 expression (Alvisi et al., **Supplementary Fig. 2B**) consistent with those seen in this study.

Thus, a combination of CD127 and CD25 expression enables the reliable isolation of Tregs from tumors.

Editorial Figure 5. CD127⁻ CD25⁺ CD4⁺ Tregs are characterized by higher FOXP3 expression compared to conventional (Tconv) and effector (Teff) CD4⁺ cells. Top: representative FACS plot for Treg identification and related FOXP3 expression. Bottom: summary of FOXP3 and additional

marker expression by CD127- CD25+ Tregs, Tconv and Teff cells. C. Bar plot showing FOXP3 MFI differences between CD127- CD25+ Tregs, Tconv and Teff cells (n=5).

15. Fig.4A: The Pearson correlation analysis revealed that CD127-CD39hi CD8 T cells were directly correlated with cells in CD4+ cluster 5 which they claim contain Treg cells. For that reason, the authors decided to look at Treg cells in the TME, more precisely at the two different subsets of Treg cells they recently identified based on IRF-4 expression (Alvisi G et al., J Clin Invest 2020) which is interesting. In that case, Fig.4A does not bring any useful information apart from suggesting that not all the cells in cluster 5 are Treg cells. It would be more appropriate to directly look at Treg cells in the tumor and determine if those two cell populations are also found in the TIL of BC patients. Based on the literature, Treg cells are believed to be negative for NKG2A and CD103. The authors should show representative flow plots demonstrating NKG2A and CD103 expression on Treg cells in their dataset (mouse Treg cells express CD103).

Author: Applying Phenograph analysis to cluster 5 of Tregs was performed to support the concept that Treg heterogeneity could be identified in an unbiased manner also in BC. We agree with the Reviewer that not all the cells in Phenograph cluster 5 may be defined as Tregs. To circumvent this issue, we applied Phenograph to manually gated CD4+ CD127- CD25+ Tregs from tumor samples as well as other tissues and visualized the resulting data by using the modified approach described previously (**Fig. 4A** - please note that clusters 5 and 7 are largely similar, and only discordant for CD39 expression, and are thus referred to as HLA-DR^{dull}). **Fig. 4A,B** show that clusters 5,7 of HLA-DR^{dull} and cluster 1 of HLA-DR^{hi} constitute the majority of Tregs in tumors. HLA-DR^{dull} and HLA-DR^{high} were respectively shown to be IRF4^{lo} CCR8^{lo} ICOS^{lo} and IRF4^{hi} CCR8^{hi} ICOS^{hi} by a subsequent staining (**Fig. 4D**). Both subsets express increased levels of FOXP3 compared to Tconv. In line with previous data (Alvisi et al., J Clin Invest, 2020), ICOS^{hi} CCR8^{hi} express more FOXP3 compared to ICOS^{lo} CCR8^{lo}.

We thank the Reviewer for raising a concern on NKG2A and CD103 expression by Tregs. As explained above, misinterpretation of the data was due to the previous visualization approach of FACS data. The new visualization approach shows that these markers are not meaningfully expressed by Tregs.

16. Fig.4B: the gate drawn to identify Treg cells shown in this figure is not appropriate. As drawn, it might include a significant number of non-Treg CD4 T cells which could bias the downstream analysis. A tighter gate around the CD127loCD25hi cells would be better. The authors should present a summary of the frequency of the markers IRF-4, CD39 and HLA-DR on Treg cells in their dataset.

Author: We thank the reviewer for the comment and we modified the gate for CD127- CD25+ Tregs in **Fig. 4C**, accordingly. The frequency of CD39 and HLA-DR by subsets of CD4+ T cells can now be appreciated in **Fig. 1B** and **Fig. 4A** thanks to the new visualization method. **Editorial Fig. 5** above shows the percentage of expression of these markers among Tconv, Teff and Treg cells. We believe these data are not essential for the message of the paper and redundant with our previous publication in J Clin Invest, and thus are not included in the main manuscript.

17. Fig.4C and D: the authors show that in the presence of a strong ICOS^{hi}CCR8^{hi} Treg signature, the positive correlation of CD127-CD39^{hi} CD8 Trm with survival is reduced in BC patients. Based on this analysis, do the authors believe that both subsets interact in the tumor? If so, are they planning to analyze the spatial location of those two subsets by IHC?

Author: the Reviewer has raised a relevant, important point. We could not address IHC staining due to the current limitations in obtaining BC tissue samples. Alternatively, we have used a computational approach, the CellPhoneDB algorithm (Roser-Vento, Nature, 2018), to calculate the receptor:ligand interactions between Tregs (scRNAseq CD4 cluster 3 from Fig. 1A) and Trm cells expressing *ENTPD1* (encoding CD39; scRNAseq CD8 cluster 4) or *IL7R^{hi}* T cells (scRNAseq CD8 cluster 1) as a comparison. The list of these interactions is shown in Editorial Table 1. Please note that the Trm engage many more interactions ($P < 0.05$) in comparison to *IL7R^{hi}* cells. Obviously, this is just a qualitative evaluation of the crosstalk between Trm and Tregs that should be investigated at the mechanistic level more in detail in a separate project. Some of these interactions involve molecules expressed by T cells (LTA, CXCL16, CCL4, CCL5) which could favor recruitment of Treg cells at their proximity. We are not including these data in the manuscript as we believe they are preliminary investigations. If the Reviewer thinks they are important, we will be happy to include them with improved graphics in the manuscript.

interacting_pair	gene_a	gene_b	Treg→IL7R CD8	Treg→CD8 Trm	IL7R CD8→Treg	CD8 Trm→Treg
TFRC_TNFSF13B	TFRC	TNFSF13B	1	0	1	1
LTA_TNFRSF1A	LTA	TNFRSF1A	1	0	1	1
LTA_TNFRSF1B	LTA	TNFRSF1B	0.16	0	1	1
CD2_CD58	CD2	CD58	1	0	1	1
LGALS9_SLC1A5	LGALS9	SLC1A5	1	0	1	1
CXCR6_CXCL16	CXCR6	CXCL16	1	0	1	1
LGALS9_SORL1	LGALS9	SORL1	0.024	0	1	1
ICAM3_aLb2 complex	ICAM3		0	0	0.324	0.06
SELL_SELPLG	SELL	SELPLG	0	0	1	1
CD74_COPA	CD74	COPA	1	0	1	1
TNFRSF1B_GRN	TNFRSF1B	GRN	1	0	0.736	0.072
CD74_MIF	CD74	MIF	0	0	1	0
TNF_TNFRSF1A	TNF	TNFRSF1A	1	0.024	1	1
ICAM2_aLb2 complex	ICAM2		0.024	0.024	1	1
LGALS9_CD47	LGALS9	CD47	0	0.024	1	1
TNF_NOTCH1	TNF	NOTCH1	1	0.036	1	1
LGALS9_CD44	LGALS9	CD44	0.52	0.048	1	1
TNF_TNFRSF1B	TNF	TNFRSF1B	0.668	0.144	1	0
KLRB1_CLEC2D	KLRB1	CLEC2D	0.08	0.684	0	0.048
CD27_CD70	CD27	CD70	1	1	0.072	0
MIF_TNFRSF14	MIF	TNFRSF14	1	1	0.096	0
TNF_ICOS	TNF	ICOS	1	1	1	0
TNF_FAS	TNF	FAS	1	1	1	0
TNFSF12_TNFRSF25	TNFSF12	TNFRSF25	1	1	1	0
CD47_SIRPG	CD47	SIRPG	1	1	0	0
TNFRSF1A_GRN	TNFRSF1A	GRN	1	1	1	0
CCL5_CCR5	CCL5	CCR5	1	1	0	0
CCL4_CCR5	CCL4	CCR5	1	1	0	0.156
CD8 receptor_LCK		LCK	1	1	0	0

Editorial Table 1. Receptor:ligand interactions identified between selected T cell clusters obtained by scRNAseq data from Figure 1A. Significant interactions are highlighted in red.

Reviewer #2

Summary

1. The purpose of these studies are to provide a better understanding of the immune milieu within breast cancers, particularly luminal-like subtypes. Compared to other BC subtypes such as TNBC, immunotherapies have failed to show efficacy in luminal-like BCs. This resistance is attributed to their immunologically cold TME compared to other subtypes. However, a better understanding of the T cell infiltrates could identify strategies by tailoring immunotherapies and could be used with CDKi which have been shown to promote tumor immunogenicity is currently used in the clinic.

In this manuscript, the authors first develop a flow cytometry panel based on re-analysis of single cell rna-seq data from previous studies. Given the recent role of tissue resident memory T cells in tumor biology, they used this panel to explore heterogeneity of CD103+CD69+ by expression of HLA-DR, CD127 and CD39. They then go on to show that these populations of functionally distinct, and that the signature of CD127-CD39+ CD8 Trm correlates with better patient outcome. Lastly, they suggest that the level of ICOS+ CD4 Tregs also impacts patient outcome.

Overall Impression

Overall the studies provide insight into the composition of T cells, particularly Trm, within BCs. However, the background and data presented is often not cohesive. For instance, the introduction focused on immunological differences between luminal-like from other BCs. However, there were few comparisons between the subtypes. There is also some concern for the absence FOXP3 expression for Treg studies. Below are other specific comments.

Authors. We understand the concern of the Reviewer. Since TNBC has been certainly studied more than luminal-like BC as far as the immune compartment is concerned, we thought we had to refer to a more general background. It has to be stressed that a deep characterization of the T cell compartment in the luminal-like BC subtype is missing, therefore little data is available to be referenced. We will modify the Introduction to include more reference to luminal-like BC literature.

Specific Comments

2. In Figure 1A, the authors show clustering of T cells using previously published datasets. It is unclear how their analytical approach differs from the original paper. Were any comparisons between the subtypes performed? In addition, was this analysis focused on luminal-like BCs given the title of the manuscript? If not, comparisons of the subtypes could have informed flow panels for the remainder of the studies.

Throughout these studies, many of the subset names and genes used to define them seem arbitrary. For instance "effector-memory like" on line 167. As well as the use of JUNB and SELL used to ascribe early CD4 memory, among others. References should be provided for how these

genes were used to annotate clusters.

Authors. Given the heterogeneity of the clinical cohort by Azizi et al, we thought that performing comparisons between patients would have been little informative, especially because Azizi et al included 8 BC patients, 4 of which were luminal-like BCs, 2 were TNBC and 1 was HER2-enriched. As we were going to analyze BC samples from a consecutive case series, mirroring BC epidemiology, we thought that the information we could gain from Azizi et al. cohort would fit our investigation. Moreover, from the total of 47,016 CD45+ cells comprised in the analysis by Azizi et al., in order to better select markers to be used to build our informative T-cell based flow cytometry panel, we had to isolate CD3+ cells (n=3,637). Thus, we might have not reached a critical number of CD3+ cells for our analysis if we had taken in consideration only luminal-like cases.

We agree with the Reviewer that definition of T cell subsets as identified by flow cytometry might be arbitrary in some cases, although it is generally based on major and widely accepted lineage markers (in the case of the “effector memory population” the reviewer is referring to, we based such definition on the basis of the CCR7⁻ CD28^{dull} CD27^{dull} phenotype, as we proposed years ago in *Mahnke et al, Eur J Immunol, 2013*). While reference gene sets are generally used to define scRNAseq clusters, a similar approach is missing for flow cytometry (we raised this issue in our recent paper by *Brummelman et al., Nat Protoc, 2019*). Unfortunately, no one to our knowledge has provided a robust approach to automatically classify T cell subsets according to major phenotypes. In any case, we will modify the text to better explain the approach we used to define T cell subsets.

3. There were also inconsistencies between the text and figure with regard to Tregs, which are denoted as C5 in the figure yet vs C4 in the text

Authors. The mistake has been corrected.

4. For the flow cytometric panel used throughout these studies, it would be helpful to provide rationale for why these markers were chosen from this analysis. For instance, a heat map with these markers could be made if they were found to be differentially expressed between the clusters.

Authors. We thank the Reviewer for pointing this aspect. Whenever possible, scRNAseq data are generally explored by our group to inform us on the identity of the T cell infiltrate in a given tissue and thus guide FACS analysis. Please note that some markers might be redundant, e.g., GZMB and GNLY. In this specific case, for instance, antibodies directed to measure the former generally give a much brighter signal compared to those measuring the latter, hence justifying the choice of one marker over the other. Otherwise, some markers may not have a good antibody (e.g., JUN) or might not be expressed ex vivo in the absence of stimulation (IFN- γ). These are general considerations and are summarized in our recent technical paper by *Brummelman et al., Nat Protoc, 2019*. We will modify the text in a revised version of the manuscript to include these considerations.

5. Representative flow cytometry plots should be more often. This includes for Figure 1B and 1C showing the gating strategy and expression markers used for heatmaps in the figures.

Authors. We apologize if we did not provide enough raw data (also requested by Reviewer 3). These are now shown in multiple instances throughout the manuscript in main and supplementary Figures.

6. For Figures 1B and 1C, it might be informative for using established gating methods to determine the frequencies of known subsets to compare between subtypes. It also doesn't look like FOXP3 was in the panel list. This seems necessary, especially in figure 4.

Authors. Also in response to Reviewers 1 and 3, we are providing quantification and related statistical analyses of the major populations of interest as identified by manual gating (**Figs. 2B, C; Figs. 4C, D; Supplementary Fig. 2B and 4B**). These data confirm results obtained by computational analyses. We understand the Reviewer's concern regarding the use of FOXP3 to identify Tregs (the same concern has been raised by Reviewer 3). Please see our comprehensive response to Reviewer 3 above to this matter and the new data provided in **Fig. 4A-D** and **Supplementary Fig. 4B**.

7. In Figure 1C, it is difficult to see if there are asterisks signifying statistical significance between the BC types. Even if none was found, the statistical analysis was used to determine this should be described as there does appear to be differences for Lum-A in C6-8.

Authors. We apologize for the inconvenience. We confirm we did not find statistically significant differences between BC subtypes (now **Fig. 1D**). The style of the figure has been changed slightly for a better comprehension by the reader.

8. Results in Figure 1C suggests no differences in the frequency of T cell subset clusters between each BC type. However, this seems contradictory to previous literature they cite in the introduction. The authors might comment or clarify whether their results differ from previous studies, if they do.

Authors. Indeed, little data is available concerning the comparison of the composition of the immune infiltrate of TNBCs and luminal-like tumors. To our knowledge, no such qualitative comparison at single-cell level has been published to date. On the other hand, there are many retrospective studies on large cohorts of patients, assessing TILs quantitative differences between TNBCs and luminal-like BCs. We know that highly proliferating tumors harbor a higher percentage of TILs, thus TNBCs, HER2 enriched and luminal B-like tumors are more infiltrated compared to luminal A-like ones (*Denkert et al. Lancet Oncol 2018; Loi et al. Ann Oncol 2014*). Specifically, only when considering chemotherapy treated TNBCs, a robust correlation between the percentage of TILs and prognosis was observed (*Adams et al. J Clin Oncol 2014; Dieci MV Ann Oncol 2014; Loi et al J Clin Oncol 2013*), making TNBCs a natural target for immunotherapeutic strategies. Nevertheless, clinical trials assessing the efficacy of immune manipulations in TNBCs have shown

conflicting results, with only a small proportion of PD-L1+ TNBCs (around 20%) demonstrating a short-term benefit in terms of disease-free survival. We will modify the Introduction to emphasize these concepts.

9. In figure 2, why were CD103+CD69+ CD8+ Trm cells divided by CD127 rather than CD39, which has been shown to indicate tumor-specific T cells in previous studies?

Authors. From the clustering data in Fig. 1B and UMAP analysis of all of the specimens in **Fig. 2A**, we saw that it was common to observe CD127 expression in the absence of CD39. Raw data of this anti-correlation is now shown in **Supplementary Fig. 2A**. Thus using one or the other marker should give nearly identical results. We apologize if this was not clear and now clarify in the text.

10. The effector function in 2C compares HLA-DR+ vs HLA-DR- in 2C/D (rather than by CD127) while 2B is on CD127 vs CD127+. The authors should consider doing 2B based on HLA-DR as well.

Authors. Please see response to the previous point as well as to Reviewer 3 regarding this matter. It is common to observe HLA-DR and CD39 co-expression in the absence of CD127 in clusters of Trm cells (**Fig. 1B**) and by UMAP analysis of single cells (**Fig. 2A**). We are confident that data reported in **Fig. 2A** enable the definition of HLA-DR+ and HLA-DR- Trm cell phenotypes in the whole cohort without the need to provide additional statistics.

11. Did the authors investigate whether there are differences between the subsets shown in Figure 2 between BC types that might explain differences in responsiveness to immunotherapy, particularly PD1?

Authors. This is an important point raised by the Reviewer. Data in **Fig. 1D** show that the relative abundance of T cell clusters is similar between different BC subtypes. We address this matter in the Discussion. Nevertheless, we acknowledge some limitations that can be mentioned in a revised version of the Discussion, should the Reviewer think are important. First, the TNBC cohort in our study is rather limited in number (we studied consecutive patients collected in a 1-year time period). Please note that characterization of the luminal-like T cell infiltrate rather than comparison with TNBC was the major goal of the study. Second, we evaluated the quality, not the quantity of the T cell infiltrate in luminal BC. This is generally considered cold, therefore it is possible that increased absolute numbers of T cells in TNBC compared to luminal BC are responsible for the preferential response to PD-1 blockade by the former. Obviously, other immune components we did not investigate (myeloid, DC, others) could also play a role in this regard.

12. Figure 2E seems out of place. Were any of the other subsets analyzed between LNs?

Authors. We believe the Reviewer refers to other subsets of BC. Unfortunately, we did not have samples that we could investigate in this regard. We slightly changed the organization of Fig. 2 and modified the text accordingly for a better comprehension by the reader.

13. In figure 3, was there a particular reason why authors chose to sort CD127+CD39^{lo} vs CD127-CD39^{hi}. For instance, why not comparisons to CD39^{+/-} or HLA-DR^{+/-}?

Authors. Justification for this choice is now included in the text at page 8. We apologize if this was not clear.

14. A better description of the signature generated for the subsets used in the survival studies should be provided. It seems from the GO analysis that many of these genes are involved in common pathways that aren't T cell-specific.
(i.e. is it possible that the survival correlation is mostly due to differences in genes expression in the cancer cells themselves between tumors)

Authors. The point raised by the Reviewer is well taken. It is true that GSEA identified gene sets that may not appear T cell specific (this occurs for any type of T cell-derived dataset in our experience), although it also identified many immune-related gene sets, including TNF, CTLA4, chemokine, IL2-STAT5 and effector T cell-related pathways. We will modify the Results section in a revised version of the manuscript to include more reference to immune-related pathways and mention the possibility raised by the Reviewer in the discussion.

15. In Figure 4a, the authors use the cluster from 1B previously thought to be Tregs for further delineation. However, the authors do not provide compelling evidence that these are indeed Tregs as FOXP3 is never shown. Although CD127-CD25^{hi} is often used to identify Tregs for sorting, FOXP3 should be used in their phenotypic studies as this is the definitive marker of Tregs, especially since intracellular IRF4 was already stained.

Authors. We thank the Reviewer for raising this concern. We provide new data showing that the vast majority of CD127- CD25⁺ Tregs isolated from tumors indeed express FOXP3. Please see our response to Reviewer 3 point 14 in this regard.

16. Again, the names for the Tregs seems arbitrary and should have citations for previous studies that have biologically defined "effector", "quiescent", etc...subtypes using these markers

Authors. We have clarified the use of nomenclature in response to Reviewer 2 point 2 above. We will modify the text in a future version of the paper, if necessary, and include related references.

17. Given the authors previous findings of ICOS and CCR8 in defining distinct Treg subsets, it is curious that they did not show these proteins in the heatmap in Figure 4.

Authors. Unfortunately, the panel to investigate T cells in BC was designed and run before we could identify subsets of Tregs in Alvisi et al., J Clin Invest, 2020. In any case, we included markers that we now know are more abundant in CCR8^{hi} ICOS^{hi} compared to CCR8^{lo} ICOS^{lo} Tregs, such as HLA-DR and, at a lesser extent, OX40. In the revised version of the manuscript, we provide additional data to formally demonstrate that HLA-DR preferentially identifies ICOS^{hi} CCR8^{hi} cells (**Fig. 4D**).

18. Were the frequencies of the Treg subsets not performed? IF so, were there differences between BC types? In addition, were there any correlation analysis performed as in 4A based on the Treg subsets.

Authors. We apologize with the reviewer if description of our results was not clear. The first version of the manuscript displayed the relative abundance of clusters of Tregs by using balloon plots and related statistics. In the new **Fig. 4A** and **B**, we propose a new visualization method that is more intuitive for the display of flow cytometry data. We show the relative abundance of specific clusters among the three different tissues analyzed, as well as their abundance related to total CD127- CD25+ Treg cells and related statistics. We can easily compute differences in Treg subsets between BC subtypes to be provided in a revised version of the manuscript, although we do not expect to identify major differences on the basis of the response to point 11 above. We did perform a Pearson's correlation between Treg and CD4+ and CD8+ T cell clusters but the results were not informative (not shown). Instead, we decided to rely on previous findings on the correlation between the absolute abundance of activated Tregs and Trm cells, defined by the use of specific signatures obtained by RNA-seq.

Reviewers' Comments:

Reviewer #1:

Remarks to the Author:

All my concerns have been addressed.

Just few details:

- regarding my comment 4, that was probably unclear: I was not saying that "the abundance of clusters 7 and 6/8 is not different between the tumor and the blood or the adjacent tissue", by that cluster 7 was not more abundant than 6/8 in the tumor.
- regarding comment 8, I apologize for not mentioning the authors' paper on T_{pex} cells. I suggest adding a brief comment about similarities and differences between T_{pex} and T_{rm} in the discussion.

Reviewer #2:

Remarks to the Author:

The authors have made considerable changes to the manuscript and have added additional data since the first submission that have largely addressed previous reviewers concerns. This includes 1. Validating FOXP3 expression by CD127-CD25+ CD4+ T cells, 2. Providing additional representative flow plots of the data used in their analysis, 3. Fixing errors in the original text and figures, and 4. Provide justification for why certain markers were used throughout their studies. These changes have improved the impact of their manuscript notably.

Minor Comments

1. It was previously brought up why the authors choose to compare CD127+ vs CD127- rather than CD39+ vs CD39- given CD39 has been shown to help identify tumor-specific T cells. In response, the authors indicate that all CD39+ cells are CD127- and that it should not change the results. However, looking at their plot provided to reviewer 1 (point 4), it appears that there are distinct CD39neg and CD39pos T cell subsets that are in the CD127- population. While it is stated that CD39 expression changes during stimulation, the authors may consider seeing if there are differences between these two CD127- populations if the flow data is available for Figure 2B. This comparison may yield interesting findings regarding late (CD127-) memory cells that are putatively tumor-specific (CD39+) and not (CD39).
2. The introduction focuses on the differences in the immunological make up of breast cancer types and states that previous studies have found higher mutation rates and TIL infiltration in TNBC compared to luminal. Yet their results show similar immune infiltrates. Unfortunately their rebuttal primarily focuses on the lack of studies correlation with survival, which was not asked. Did the authors find differences in bulk T cell or CD45+ frequencies between tumor types? If not, the authors should comment on in their discussion why (e.g. technique, number of samples) there may be discrepancies.
3. As stated previously, differences in PD1 expression between subtypes may help explain differences in responsiveness between BC types. Although in their rebuttal the authors point out that the frequency of T cell clusters are relatively the same, it could be interesting if there are differences in the percent of CD8 T cells that express PD1 between BC types.

Reviewer #3:

Remarks to the Author:

First, I would like to thank the authors for addressing most of the points I raised during the review of their manuscript. More specifically, the modified visualization approach in Fig. 1B and Fig. 4A is a great improvement over the previous heatmaps. It is easier to read and is much more informative. Moreover, the UMAPs of concatenated CD4 and CD8 are useful to compare the distribution of the different cell clusters between the blood, normal tissue and tumor. I also appreciate that they are now representing MFI instead of percentages for the different markers

they looked at in Fig. 2B. It is more relevant, especially for markers like PD-1 and HLA-DR. Finally, representing cytokine production as pseudocolor plots in Fig. 2D help better visualize differences between the two cell subsets. Altogether, the new version of the manuscript has been improved.

However, I still have some concerns that would need to be addressed:

1. The new title is more representative of the data presented in this manuscript. However, I think it would be important to add the word "CD8": "CD39hi tissue resident memory CD8 T cells" as the CD4 Trm also exist but were not the focus of this manuscript.

2. To highlight the results from Fig. 2C, the authors show representative flow plots from one BC patient. However, based on their frequency analysis, the plot for non-metastatic LN does not seem correct. In their summary, the maximum frequency in non-metastatic LN is less than 0.25% of total CD8 T cells whereas in the example they provide it is 1.45%. Please replace with correct flow plot.

In line 707, the figure legend for Fig. 2C states that the mean +/- SEM of frequency of CD127-CD39hi among total CD8+CD69+CD103+ T cells is displayed. However, I think it is not correct and that the figure instead shows the mean +/- frequency of CD127-CD39hi among total CD8+ T cells. There is the same issue for Suppl Fig.2B. It needs to be corrected.

Also, it would be useful to the readers to understand if the frequency of CD127-CD39hi CD8 Trm correlates between primary tumor and metastatic LN.

3. For the cytokine production analysis, the authors decided to use HLA-DR to separate the CD8 Trm subsets instead of CD127 and CD39. I understand that this decision was motivated by the fact the CD39 expression can be affected by PMA and ionomycin and I thank the authors for the Editorial Figure 1 which shows that HLA-DR tend to be expressed more on CD39hiCD127- than on CD39-CD127+. However, I think it would be more appropriate to first gate on HLA-DR+ cells among CD8+CD69+CD103+ and then look at the expression of CD127 and CD39. By doing so the readers would have a better idea of the enrichment of CD39hiCD127- in the HLA-DR+ vs the HLA-DR- subset. It would be useful to report the enrichment for the 8 patients that were analyzed for cytokine production.

4. The summary of the frequency of the 2 subsets of CD8 Trm in the blood, normal tissue and tumor presented in Suppl. Fig. 2B is very informative and should be added to Fig. 2.

5. In Fig. 3C, the authors represent the difference in OS for all BC patients based on the CD127-CD39hi and CD127+CD39- Trm gene signature. It would also be important to show the Kaplan Meier plot for Luminal-like BC patients only as this is the main focus of the manuscript. How does the survival plot look like when integrating CD8A expression with CD127-CD39hi gene signature (CD8Ahi CD127-CD39hi Trm High, CD8Ahi CD127-CD39hi Trm Low, CD8Alow CD127-CD39hi Trm High and CD8Alow CD127-CD39hi Trm Low)? It is an important point as the CD127-CD39hi Trm population represents a very small proportion of TIL CD8 in luminal-like BC.

6. I appreciate that the authors modified their analysis and gated on CD4+CD25+CD127- before re-clustering by Phenograph as it helps remove non-Treg cells from the analysis. I know that CD25 and CD127 are widely used to identify Treg cells in humans. However, it is still preferable to keep this gating strategy for functional assays. For phenotypic analysis, FOXP3 staining is a much better solution as it is the definitive marker for Treg cells, and its staining is very clear in human tumors. The editorial Fig. 5 shows that the authors strongly enriched in Treg cells by using CD25 and CD127 gating strategy which is fine and should be sufficient for this analysis.

The summary of FOXP3 expression (percentage and MFI) as well as the expression of CD39, HLA-DR and IRF4 should be included in Fig. 4 to support results in Fig. 4C and D as there is a lot of heterogeneity among patients and it is important to show that the data are consistent across patients.

Also, because the authors did not have CCR8 and ICOS in their original 27-parameters panel, it would make more sense in the text to first mention Treg heterogeneity using their Phenograph analysis, then focus on HLA-DR+ and HLA-DR- Treg cells and finally introduce CCR8 and ICOS in reference to their previous publication. In that regard, the sentence in lines 271 and 272 is not correct as it does not support data in Fig. 4D. In this figure, the authors gated on CCR8 and ICOS

and then looked at the expression of other markers. Instead, the text indicates that the authors gated on HLA-DR+ Treg cells and then analyzed CCR8, ICOS and IRF4 expression (which would be the right way to do it).

Providing a summary of the frequency of CCR8+ICOS+ and CCR8-ICOS- Treg subsets in BC would be necessary to the readers to appreciate the heterogeneity between patients.

Also, to support the Pearson correlation analysis, it seems important to provide a graph plotting the frequency of CD127-CD39hi Trm against total Treg cells and another one with CD127-CD39hi Trm against CCR8+ICOS+Treg cells.

7. In line 288 and 289, the authors indicate "we dissected the BC T cell immune milieu by single-cell technologies and focused on CD8+ TILs to gain deep insight on tumor-reactive,...". There is no data in this manuscript supporting tumor-reactivity of the CD127-CD39hi Trm cells so this sentence needs to be modified.

8. In line 320, the authors indicate that "the abundance of CD39hi Trm directly correlated with that of Treg cells". The data presented in this manuscript do not exactly support that. Plotting the frequency of CD127-CD39hi Trm against the frequency of total Treg cells would support (or not) this statement.

9. In line 159, the authors indicate that using Phenograph, they identified 11 clusters of CD8 T cells. However, Fig. 1B and Suppl Fig. 1A and B show only 10 clusters for CD8 T cells. Please modify in the text.

10. I am not an expert in using gene signatures to predict patient survival but I feel like the authors should provide more information about how they performed their analysis so that anyone with the proper computer skills could reproduce it.

11. I agree with the authors that their interaction analysis presented in Editorial Table 1 is still preliminary and should not be included in the manuscript. Multiplex IHC or other spatial localization techniques should be used to address that question.

12. An important take home message from that manuscript is the low frequency CD127-CD39hi Trm among total CD8 TILs. It would be important to address that point in the discussion. One goal to increase BC patients' response to immunotherapy might be to expand those cells in situ in addition to targeting Treg cells.

Point-by-point response to the Reviewers

The authors would like to thank the Reviewers for the valuable comments. We think that the manuscript has greatly improved in its revised form. We hope our changes are acceptable.

Reviewer 1

1. Regarding my comment 4, that was probably unclear: I was not saying that "the abundance of clusters 7 and 6/8 is not different between the tumor and the blood or the adjacent tissue", by that cluster 7 was not more abundant than 6/8 in the tumor.

Author: We apologize for this misunderstanding. We modified our conclusion at line 202 to incorporate the suggestion of the Reviewer.

2. Regarding comment 8, I apologize for not mentioning the authors' paper on T_{pex} cells. I suggest adding a brief comment about similarities and differences between T_{pex} and T_{rm} in the discussion.

Author: We thank the Reviewer for the comment. We added a statement in the Discussion.

Reviewer 2

1. It was previously brought up why the authors choose to compare CD127⁺ vs CD127⁻ rather than CD39⁺ vs CD39⁻ given CD39 has been shown to help identify tumor-specific T cells. In response, the authors indicate that all CD39⁺ cells are CD127⁻ and that it should not change the results. However, looking at their plot provided to reviewer 1 (point 4), it appears that there are distinct CD39^{neg} and CD39^{pos} T cell subsets that are in the CD127⁻ population. While it is stated that CD39 expression changes during stimulation, the authors may consider seeing if there are differences between these two CD127⁻ populations if the flow data is available for Figure 2B. This comparison may yield interesting findings regarding late (CD127⁻) memory cells that are putatively tumor-specific (CD39⁺) and not (CD39⁻).

Author: Please note that our initial analytical approach focusing on CD127⁺ vs CD39^{hi} T_{rm} subsets should not be intended as a lack of interest on additional, possible heterogeneity, rather as a way to simplify the identification of populations that are relevant in biological processes. We certainly understand the Reviewer's request to look deeper in CD127⁻ cell heterogeneity by standard gating of flow cytometry data to possibly identify additional interesting information. Among these CD127⁻ cells, we could indeed recognize one subset of CD127⁻ CD39^{hi} cells and one subset of CD127⁻ CD39^{lo} cells. We updated Figure 2B on manual gating analysis of marker expression by T_{rm} subsets identified by combinatorial expression of CD127 and CD39. CD127⁻ CD39^{hi} cells are preferentially HLA-DR⁺, PD-1⁺ and, at a lesser extent, TIGIT⁺ compared to CD127⁻ CD39^{lo} cells (in line with putative antigen recognition sustaining the expression of inhibitory receptors and activation markers; Simoni et al Nature 2018). While there is no significant difference in GZMB expression, CD127⁻ CD39^{lo} cells seem to express more GZMK. We modified the text to include description of these new data. In light of these data, CD127⁻ CD39^{lo} may represent a stage of differentiation that is intermediate between CD127⁺ CD39^{lo} and CD127⁻ CD39^{hi}. This is briefly mentioned in the Discussion at line 308, however we would avoid to draw definitive conclusions

on the functional capacity of these cells or on their lineage relationships in the absence of more in depth molecular or functional analyses, which in the authors' opinion should be the focus of a more specific project.

2. The introduction focuses on the differences in the immunological make up of breast cancer types and states that previous studies have found higher mutation rates and TIL infiltration in TNBC compared to luminal. Yet their results show similar immune infiltrates. Unfortunately their rebuttal primarily focuses on the lack of studies correlation with survival, which was not asked. Did the authors find differences in bulk T cell or CD45+ frequencies between tumor types? If not, the authors should comment on in their discussion why (e.g. technique, number of samples) there may be discrepancies.

Author: We thank the reviewer for the comment and we apologize if our previous answer was unclear on this point. Certainly the focus of the manuscript is to provide a qualitative characterization of the immune infiltrate in the luminal-like BC, rather than a comparison with other subtypes such as TNBC. As we have analyzed TNBC cases as well, we thought it was of interest to show those data. We have shortened the Introduction as we acknowledge that some clinical information regarding the differential response to checkpoint blockade immunotherapy might not be necessary for the main message of the paper.

As for the analysis of the quantity of the CD45+ immune infiltrate, it is established that TNBCs commonly show higher levels of TILs compared to the luminal-like type (Loi S et al Ann Oncol 2014), thus investigating/replicating these data in a smaller cohort of patients like the one tested here would be incremental. The size of our cohort would probably be insufficient to answer this question. Moreover, please note that a dedicated approach in this regard that follows the International TIL Working Group recommendations (Salgado R et al, Ann Oncol 2015), such as immunohistochemistry of TILs rather than their quantification by flow cytometry, would be more appropriate.

3. As stated previously, differences in PD1 expression between subtypes may help explain differences in responsiveness between BC types. Although in their rebuttal the authors point out that the frequency of T cell clusters are relatively the same, it could be interesting if there are differences in the percent of CD8 T cells that express PD1 between BC types.

Author: We thank the reviewer for the interesting comment. Evidence from clinical trials has shown that expression of PD-L1 both on tumor cells and immune cells is the only marker significantly correlating with therapeutic efficacy of immunotherapy in TNBC. Some previous study has reported a higher frequency of PD1 expression, evaluated by immunohistochemistry, in TNBCs and HER2+ tumors compared to luminal-like cases (Muenst S, Breast Cancer Research and Treatment, 2013; Zhou, Anti-Cancer Drugs, 2018; Kim H, BMC Cancer, 2017), while some other studies could not find a statistically significant difference between different biological subtypes in terms of PD-1 and PD-L1 expression (Tsang JYS, Breast Cancer Research and Treatment, 2017; Yuan C, Scientific reports, 2019). In the Editorial Figure 1 below, we provide quantification of PD-1 expression in CD4+ and CD8+ T cells among the different BC subtypes, showing there is lower expression among CD8+, but not CD4+ T cells in luminal B compared to luminal A and TNBC. However, we would avoid to show these data (the results are difficult to interpret in the absence of more specific experiments) and to speculate on the levels of PD-1 and response to

immunotherapy in the absence of a dedicated clinical trial designed to answer this specific question.

Editorial Figure 1: Bar plot showing percent expression of PD-1 in CD8+ and CD4+ cells among different BC biological subtypes (n=54 patients). *, $P < 0.05$; 1-way ANOVA with Bonferroni post-hoc test.

Reviewer 3

1. The new title is more representative of the data presented in this manuscript. However, I think it would be important to add the word “CD8” : “CD39^{hi} tissue resident memory CD8 T cells” as the CD4 Trm also exist but were not the focus of this manuscript.

Author: We thank the reviewer for the suggestion and we changed the title accordingly.

2. To highlight the results from Fig. 2C, the authors show representative flow plots from one BC patient. However, based on their frequency analysis, the plot for non-metastatic LN does not seem correct. In their summary, the maximum frequency in non-metastatic LN is less than 0.25% of total CD8 T cells whereas in the example they provide it is 1.45%. Please replace with correct flow plot. In line 707, the figure legend for Fig. 2C states that the mean +/- SEM of frequency of CD127- CD39^{hi} among total CD8+CD69+CD103+ T cells is displayed. However, I think it is not correct and that the figure instead shows the mean +/- frequency of CD127- CD39^{hi} among total CD8+ T cells. There is the same issue for Suppl Fig.2B. It needs to be corrected. Also, it would be useful to the readers to understand if the frequency of CD127- CD39^{hi} CD8 Trm correlates between primary tumor and metastatic LN.

Author: We thank the reviewer for the comment and we apologize for the mistake in Fig. 2 legend. We modified the FACS plot in Fig. S2, showing a more representative patient and we corrected the figures legend for Fig. 2 and Fig. S2. Unfortunately, we collected tumor tissues for a minority of the lymph node donors, thus we do not have matched data for the frequency of CD127-CD39^{hi} CD8+ Trm in tumor and lymph nodes to run a correlation analysis.

3. For the cytokine production analysis, the authors decided to use HLA-DR to separate the CD8 Trm subsets instead of CD127 and CD39. I understand that this decision was motivated by the fact the CD39 expression can be affected by PMA and ionomycin and I thank the authors for the Editorial Figure 1 which shows that HLA-DR tend to be expressed more on CD39^{hi}CD127- than on

CD39-CD127+. However, I think it would be more appropriate to first gate on HLA-DR+ cells among CD8+CD69+CD103+ and then look at the expression of CD127 and CD39. By doing so the readers would have a better idea of the enrichment of CD39^{hi}CD127⁻ in the HLA-DR+ vs the HLA-DR⁻ subset. It would be useful to report the enrichment for the 8 patients that were analyzed for cytokine production.

Author: We have previously stated that CD127 and CD39 undergo dynamic changes in response to this type of stimulus. These phenotypic changes are generally observed in some, but not all the individuals: from here the idea to use surrogate markers (i.e., HLA-DR, as inspired by high-dimensional single cell profiling, Fig. 1) as a proxy for the identification of the Trm subsets. We understand that this might not be the optimal situation due to the non-complete overlap between marker expression. Moreover, Reviewer 2 asked to look more in detail in the characteristics of the 3 Trm subsets identified by CD127 and CD39 (please see our detailed response to Reviewer 2, point 1 on this matter). To be more consistent with the presentation of the data and to avoid the use of the surrogate marker HLA-DR for the identification of the different Trm subsets, we had the possibility to collect more samples (or to use archival samples), stimulate them with PMA/ionomycin and include in the analysis only those samples that maintained an immunophenotypic profile equivalent to that of the unstimulated condition (n=6 patients). We still do not know why some individuals change their phenotype while others don't in response to PMA/ionomycin, but this observation has been consistent in a number of different studies we have published in the past (including *Roberto et al., Blood, 2015*; *Galletti et al., Nature Immunol, 2020*; *Bonnal et al., Nature Immunol, 2021*). Thus, we have restructured Figure 2 where we are showing cytokine production profiles of the 3 subsets identified by CD127 and CD39 expression. We are now including also profiles of IL-2 production by these subsets, as we generated a novel flow cytometry panel including a very bright anti-IL-2 reagent. While the 3 subsets produced equal amount of TNF and IFN- γ , we detected subset-specific differences regarding IL-2 production, which is highest in CD127⁺ CD39^{lo}, intermediate in CD127⁻ CD39^{lo} and low in CD127⁻ CD39^{hi}. In line with our previous results, we have found increased degranulation capacity (CD107a production) in CD127⁻ CD39^{hi} compared to CD127⁺ CD39^{lo}, reflecting their higher GZMB content ex vivo. CD107a production (and GZMB content) in CD127⁻ CD39^{lo} cells did not differ from that of CD127⁻ CD39^{hi}. In light of these new data, we slightly rephrased the description and the interpretation of the results, and we refer to CD127⁻ CD39^{hi} cells as those with increased degranulation capacity only when compared to CD127⁺ CD39^{lo} Trm.

4. The summary of the frequency of the 2 subsets of CD8 Trm in the blood, normal tissue and tumor presented in Suppl. Fig. 2B is very informative and should be added to Fig. 2.

Author: We thank the reviewer and we added Figure 2C in which we show the enrichment in tumor, normal tissue and blood of the three Trm subpopulations characterized in Figure 2B.

5. In Fig. 3C, the authors represent the difference in OS for all BC patients based on the CD127-CD39^{hi} and CD127⁺CD39⁻ Trm gene signature. It would also be important to show the Kaplan Meier plot for Luminal-like BC patients only as this is the main focus of the manuscript. How does the survival plot look like when integrating CD8A expression with CD127-CD39^{hi} gene signature (CD8A^{hi} CD127-CD39^{hi} Trm High, CD8A^{hi} CD127-CD39^{hi} Trm Low, CD8A^{low} CD127-CD39^{hi} Trm

High and CD8A low CD127-CD39^{hi} Trm Low)? It is an important point as the CD127-CD39^{hi} Trm population represents a very small proportion of TIL CD8 in luminal-like BC.

Author: We thank the reviewer for raising this important aspect. In Figure 3D we show Kaplan-Meier curves for OS, in luminal-like tumors extracted from METABRIC dataset, for the enrichment of different Trm subsets. This is now specified in the text and the figure legend. Consistent with our previous data, the CD127- CD39^{hi} Trm signature positively modulate prognosis ($p < 0.0001$), while CD127+ CD39^{lo} Trm signature does not.

We understand the importance of the Reviewer's request of integrating the survival analysis with CD8A levels. In a **new Supplementary Figure S3D** we show Kaplan Meyer curves for OS analyzing interaction of CD8A expression with the two different Trm subpopulations signatures. As expected, CD127- CD39^{hi} Trm signature mediate better prognosis when CD8A is highly expressed, the opposite is seen when levels of both are low. Patients with CD127- CD39^{hi} Trm High CD8A low seem to do slightly better than CD127- CD39^{hi} Trm low CD8A high. As expected, patients with CD127- CD39^{hi} Trm low CD8A low have the worst prognosis. These data are also discussed.

New Supplementary Figure S3D: *Kaplan-Meier overall survival (OS) curves in the METABRIC consortium data set in luminal-like BCs (n=1,436) according to high or low enrichment of CD127-CD39^{hi} Trm, integrated with CD8A expression. The mean z-score value was used to classify tumor samples into LOW and HIGH expression groups. P values were calculated applying the Log-rank (Mantel-Cox) test.*

6. I appreciate that the authors modified their analysis and gated on CD4+CD25+CD127- before re-clustering by Phenograph as it helps remove non-Treg cells from the analysis. I know that CD25 and CD127 are widely used to identify Treg cells in humans. However, it is still preferable to keep this gating strategy for functional assays. For phenotypic analysis, FOXP3 staining is a much better solution as it is the definitive marker for Treg cells, and its staining is very clear in human tumors. The editorial Fig. 5 shows that the authors strongly enriched in Treg cells by using CD25 and CD127 gating strategy which is fine and should be sufficient for this analysis.

The summary of FOXP3 expression (percentage and MFI) as well as the expression of CD39, HLA-DR and IRF4 should be included in Fig. 4 to support results in Fig. 4C and D as there is a lot of heterogeneity among patients and it is important to show that the data are consistent across patients.

Also, because the authors did not have CCR8 and ICOS in their original 27-parameters panel, it would make more sense in the text to first mention Treg heterogeneity using their Phenograph analysis, then focus on HLA-DR+ and HLA-DR- Treg cells and finally introduce CCR8 and ICOS in

reference to their previous publication. In that regard, the sentence in lines 271 and 272 is not correct as it does not support data in Fig. 4D. In this figure, the authors gated on CCR8 and ICOS and then looked at the expression of other markers. Instead, the text indicates that the authors gated on HLA-DR+ Treg cells and then analyzed CCR8, ICOS and IRF4 expression (which would be the right way to do it).

Providing a summary of the frequency of CCR8+ICOS+ and CCR8-ICOS- Treg subsets in BC would be necessary to the readers to appreciate the heterogeneity between patients.

Also, to support the Pearson correlation analysis, it seems important to provide a graph plotting the frequency of CD127-CD39^{hi} Trm against total Treg cells and another one with CD127-CD39^{hi} Trm against CCR8+ICOS+Treg cells.

Author: We thank the Reviewer for acknowledging the importance of the new data regarding the phenotypic profile of tumor Tregs vs. conventional (Tconv) and effector (Teff) cells. We are now including graphs of the summary of FOXP3, CD39, HLA-DR and IRF4 expression in a **new Fig. 4D** (please note we updated the figure due to a mistake we have found in a previous version – conclusions remain unchanged). We have restructured the text to follow the Reviewer’s instructions on how to present the results (first mentioning Treg heterogeneity by Phenograph, then introducing HLA-DR expression by the Treg, then reference to our previous J Clin Invest paper on CCR8 and ICOS expression by HLA-DR+ Tregs). We have also changed the gating strategy in **Fig. 4E** as suggested by the Reviewer. The phenotypic difference between HLA-DR- and HLA-DR+ Tregs in the tumor is consistent among different donors, although a summary is not shown because we thought it would be redundant with data in Fig. 4D asked by the Reviewer. Our previous conclusions remain unchanged.

Please also note that differences in CCR8, ICOS and IRF4 expression between these two Treg populations may not be as clear cut as one would expect (i.e., completely negative or completely positive – indeed, also in our previous Alvisi et al., J Clin Invest, 2020 publication we have shown that quiescent Tregs in tumors express some levels of CCR8 and ICOS). In any case, the half-a-log to 1-log differences in these marker expression is shown to justify the use of the activated ICOS^{hi} Treg signature from a previous RNA-seq experiment.

In a **new Supplementary Fig. 4B** we provide the graph for specific correlation between the frequency of CD127-CD39^{hi} Trm and total Treg cells ($p=0.0011$), as requested by the Reviewer. A summary of the frequency of CCR8^{hi} ICOS^{hi} Treg subset is also shown along with the related gate in **Supplementary Fig. 4D**. The specific correlation between CD127-CD39^{hi} Trm and CCR8^{hi} ICOS^{hi} Treg cells ($p=0.0529$, showing a trend toward significance) is provided below in Editorial Fig. 3. Unfortunately, we do not have that many patients’ samples to be included in such correlation and could not enroll more due to the global pandemic situation, therefore we would prefer to not include these data in the main manuscript.

Editorial Figure 3: specific correlation between CD127-CD39^{hi} Trm and CCR8^{hi}ICOS^{hi}Treg cells (Pearson $r=0.8059$, $p=0.0529$).

7. In line 288 and 289, the authors indicate “we dissected the BC T cell immune milieu by single-cell technologies and focused on CD8+ TILs to gain deep insight on tumor-reactive,...”. There is no data in this manuscript supporting tumor-reactivity of the CD127-CD39^{hi} Trm cells so this sentence needs to be modified.

Author: we have modified the sentence by removing the reference to tumor reactivity of Trm cells

8. In line 320, the authors indicate that “the abundance of CD39^{hi} Trm directly correlated with that of Treg cells”. The data presented in this manuscript do not exactly support that. Plotting the frequency of CD127-CD39^{hi} Trm against the frequency of total Treg cells would support (or not) this statement.

Author: This conclusion is now supported by manually gated flow cytometry data presented in the **new Supplementary Fig. 4B**.

9. In line 159, the authors indicate that using Phenograph, they identified 11 clusters of CD8 T cells. However, Fig. 1B and Suppl Fig. 1A and B show only 10 clusters for CD8 T cells. Please modify in the text.

Author: We thank the reviewer for the comment and we apologize if we did not clearly specify that clusters with a frequency <1% were excluded from the analysis to simplify visualization. Thus, 11 clusters were identified by unsupervised clustering algorithm, but only 10 were considered for analysis. We now specify this in the text and in Methods.

10. I am not an expert in using gene signatures to predict patient survival but I feel like the authors should provide more information about how they performed their analysis so that anyone with the proper computer skills could reproduce it.

Author: We thank the reviewer for the comment, we better specified in the Methods section regarding Survival analysis how normalized signal values for each gene expression signatures were converted into z-scores and the formula used.

11. I agree with the authors that their interaction analysis presented in Editorial Table 1 is still preliminary and should not be included in the manuscript. Multiplex IHC or other spatial localization techniques should be used to address that question.

Author: We totally agree with the reviewer on the potential importance of using spatial localization techniques in order to better and deeply understand cell-to-cell interactions, but unfortunately, due to the current limitations in obtaining adequate BC tissue samples, we could

Losurdo et al.

not address this specific point. We surely would like to do in the future, in the prospective follow-up of our work.

12. An important take home message from that manuscript is the low frequency CD127-CD39hi Trm among total CD8 TILs. It would be important to address that point in the discussion. One goal to increase BC patients' response to immunotherapy might be to expand those cells in situ in addition to targeting Treg cells.

Author: We have expanded the Discussion to include this specific point.

Reviewers' Comments:

Reviewer #3:

Remarks to the Author:

In their revision, Losurdo et al. have significantly improved their manuscript, addressing almost all concerns with their additional analyses and figure changes. Altogether, this new version of the manuscript is clearer and stronger.

Losurdo et al.

Point-by-point response to the Reviewers

The authors would like to thank the Reviewers for the valuable comments.